# From Retrieval to Translation: Translating Query into Graph-level Clues for Retrieval-Augmented Generation

Qichuan Liu [* 1]   Qinggang Zhang [* 2]   Yuxuan Hu [1]   Chenfeng Zheng [1]   Zerui Chen [1]   Chentao Zhang [1]
Zhihong Zhang [1]

## Abstract

Retrieval-Augmented Generation (RAG) has recently been enhanced with tree or graph structures to match user intent for precise passage retrieval, which facilitates large language models (LLMs) in effectively mitigating hallucinations by leveraging external knowledge. However, we identify that existing structure-augmented RAG systems are experiencing (i) *potential retrieval suspension* and (ii) *cumulative semantic drift*, due to low-quality structures and semantic embeddings that often poorly capture textual details. Motivated by this, we propose a novel paradigm named **KG-Translator**, which is distinct from traditional matching-based paradigms and instead translates user queries into graph-level clues. Specifically, KG-Translator utilizes lightweight models to conduct named entity recognition (NER) and syntactic parsing on the corpus, constructing a reliable knowledge graph (ParseKG). On top of ParseKG, KG-Translator adopts constrained decoding strategies to faithfully translate clues, traces them to original passages, and employs a lightweight ranking model for precise passage retrieval. Extensive experiments on five datasets demonstrate that KG-Translator significantly outperforms baselines[1].

## 1. Introduction

Retrieval-Augmented Generation (RAG) delivers a promising approach for large language models (LLMs) to reduce hallucinations by leveraging external knowledge (Gao et al., 2023; Liu et al., 2025; Zhang et al., 2025c). However, tradi-

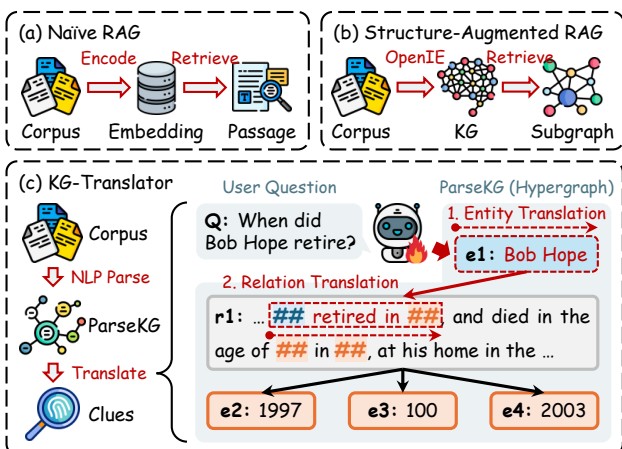

*Figure 1.* **Three paradigms of RAG systems.**

tional RAG systems face challenges in large-scale unstructured corpora: retrieval overly relies on embedding-based semantic matching, which tends to lose critical contextual details. Although recent efforts focus on large embedding models, which are post-trained on LLMs to achieve more expressive semantic embeddings (Li et al., 2023b; Muennighoff et al., 2024; Lee et al., 2025), they still fail to break free from the limitations of traditional RAG systems, resulting in insufficient or ineffective context, impairing retrieval and reasoning for complex tasks (Zhang et al., 2025b).

Recently, structure-augmented RAG has emerged as an effective paradigm that employs structured trees or graphs to model corpora and queries for fine-grained knowledge retrieval (Sarthi et al., 2024; Gutiérrez et al., 2025; Liu et al., 2025). Existing methods can be roughly categorized into two levels: *chunk-level* and *entity-level*. (1) Chunk-level methods, such as KGP (Wang et al., 2024) and RAPTOR (Sarthi et al., 2024), build indexes based on semantic similarity, and achieve knowledge organization or retrieval through LLM-based traversal; (2) Entity-level methods extract entities and relations via open information extraction (OpenIE) to construct knowledge graphs (KGs), and then perform subgraph retrieval to match target knowledge. For instance, the early work LightRAG (Guo et al., 2025) designs a dual-level retrieval paradigm to capture both fine-grained structural details and high-level semantic insights. Later, GFM-RAG

*Equal contribution [1]School of Informatics, Xiamen University, China [2]School of Artificial Intelligence, Jilin University, China. Correspondence to: Zhihong Zhang <zhihong@xmu.edu.cn>.

*Proceedings of the $43^{rd}$ International Conference on Machine Learning*, Seoul, South Korea. PMLR 306, 2026. Copyright 2026 by the author(s).

[1]Code and data are available at: https://github.com/GenIRAG/KG-Translator.

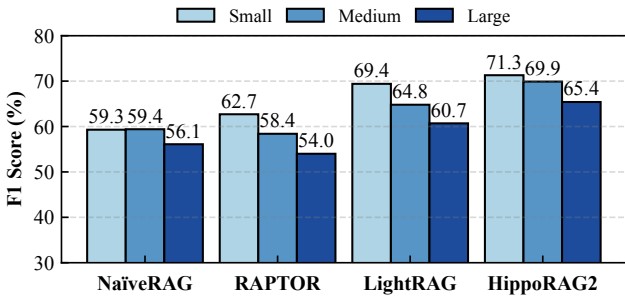

*(a)* QA performance of structure-augmented RAG with corpus scaling. Corpus sizes: Small (5k), Medium (10k), and Large (35k).

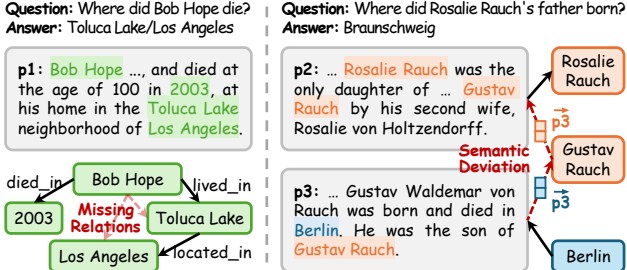

*(b)* Two error types brought by structures and embeddings: potential retrieval suspension (left) and cumulative semantic drift (right).

*Figure 2.* **(a) QA performance of structure-augmented RAG with corpus scaling.** The evaluation on HotpotQA measures F1 score (%). **(b) Case study of two error types.** The analysis is conducted from structural and embedding perspectives.

(Luo et al., 2025c) leverages query-dependent graph neural networks (GNNs) to capture inter-entity relations. More recently, HippoRAG2 (Gutiérrez et al., 2025) further extends HippoRAG (Gutiérrez et al., 2024) by leveraging personalized PageRank (PPR) to facilitate cross-passage retrieval.

Despite recent advances, structure-augmented RAG still suffers from low-quality structural modeling, which turns its structural advantages into performance bottlenecks (Zhuang et al., 2026; Chen et al., 2026). Although structural modeling enables more granular retrieval, it is often accompanied by introducing inaccurate or irrelevant information, making it difficult to ensure the accuracy of retrieval results. Furthermore, we identify the performance bottlenecks of existing structure-augmented RAG systems as shown in Figure 2. This in-depth analysis highlights two key challenges: ❶ **Potential retrieval suspension:** Relation absence in the KG leads to the interruption of search paths, preventing retrieval from reaching the target knowledge; ❷ **Cumulative semantic drift:** existing retrieval strategies rely on semantic embeddings to match queries with structures, constrained by insufficient textual detail representation of embedding models and noise from low-quality graphs. For example, for the query "Where did Rosalie Rauch's father born?" in Figure 2b (right), the embedding of $p_3$ incorrectly raises its relevance to the query due to the association between "Gus-

tav Rauch" and $p_2$, thereby inducing semantic deviation.

To tackle these issues, we propose a novel paradigm named **KG-Translator**, which transforms the retrieval task into a translation task, leveraging the strong semantic understanding capability of LLMs to translate user intent into graph-level clues for precise retrieval. Specifically, (1) Parsing-based KG Construction: A lightweight model is first employed for named entity recognition (NER) and syntactic parsing. Then, we mask all entities in sentences to obtain hyperrelations and construct a reliable graph (ParseKG). (2) Structure-constrained KG Translation: A specialized LLM leverages dynamic KG-constrained indexing to constrain the decoding process, translating user queries into graph-level clues faithfully. (3) Clue-traced Passage Retrieval: We backtrack these clues to the original passages and adopt a lightweight model for ranking, achieving precise passage retrieval. Our contributions are summarized as follows:

- We identify that existing structure-augmented RAG systems are currently facing two key challenges: *potential retrieval suspension* and *cumulative semantic drift*, which motivates the design of our KG-Translator.

- KG-Translator constructs a novel graph (ParseKG) by leveraging a lightweight model for entity extraction and syntactic parsing, cutting the graph construction time while preserving the complete passage semantics.

- On top of ParseKG, a specialized LLM leverages the KG-constrained index to faithfully translate query into clues. The clues are traced to original passages, and a lightweight ranking model delivers precise retrieval.

- We conduct extensive experiments across five benchmark datasets, demonstrating that KG-Translator consistently outperforms state-of-the-art baselines, validating its practicality for real-world applications.

## 2. Preliminary Study

Before diving into technical details, we conduct preliminary studies to investigate the limitations of structure-augmented RAG systems, revealing the structural and semantic flaws that induce the performance bottlenecks of such systems. We systematically organize the related work in Appendix B.

### 2.1. Performance Degrades with Corpus Scaling

As shown in Figure 2a, we conduct experiments on HotpotQA (Yang et al., 2018) across three corpus scales (5k, 10k, and 35k), and conclude the following observations:

**Obs. 1. Structure-augmented RAG experiences performance degradation as the corpus scales up.** As the corpus scales up from 5k to 10k, the F1 scores of RAPTOR, LightRAG, and HippoRAG2 decrease by 4.3%, 4.6%, and 1.4%,

respectively. As the scale further expands to 35k, the F1 scores of these three models still show a significant downward trend, with an average reduction of 4.3%. These results verify that structure-augmented RAG systems across different architectures all experience performance degradation as the corpus scale increases. More importantly, this phenomenon highlights a core issue: the fine-grained semantic association mechanism of structure-augmented RAG may suffer from systematic interference during corpus scaling.

**Obs. 2. The degradation rate of structure-augmented RAG is steeper than the rate of NaïveRAG.** Across two corpus expansion phases, structure-augmented RAG systems exhibit average F1 score declines of 3.4% and 4.3%, whereas NaïveRAG has a 0.1% rise and a 3.3% drop. The former has a performance gap of 7.7%, versus 3.2% for the latter. This highlights the scale sensitivity of the structure-augmented RAG in contrast to the stronger adaptability of NaïveRAG. Additionally, NaïveRAG surpasses RAPTOR by 2.1% after the second expansion, proving that structural benefits of RAPTOR hinder performance gains. Overall, gains from structure depend on balancing association accuracy and scale robustness. Section 2.2 further examines error cases to reveal the root causes of performance degradation.

### 2.2. Error Analysis and Discussion

To identify the core cause of the performance degradation observed in Figure 2a, we conduct an error analysis of the structure-augmented RAG systems, indicating that these errors derive from the structural and semantic flaws.

**Case 1: Potential retrieval suspension.** As OpenIE, especially relation extraction, tends to overlook implicit knowledge associations in text (Zhuang et al., 2026), it gives rise to structural flaws in the KG. This disrupts critical search paths during retrieval, thereby leading to retrieval direction deviation. For example, in the "Bob Hope" case (Figure 2b, left), though the text mentions that he died at his home in Toluca Lake, OpenIE failed to extract the "died_in" relation between Bob Hope and Toluca Lake during extraction.

**Discussion 1.** Large-scale corpora amplify the structural flaws of structure-augmented RAG, directly impairing retrieval quality and ultimately degrading performance. We attribute this to the following: (1) The instability of LLM generation: performing OpenIE requires well-designed prompts, yet still fails to guarantee the comprehensiveness of extraction, resulting in poor quality of the constructed graphs; (2) The oversimplification of relations: relations are often represented as concise expressions (e.g., "is_a"), which cannot fully cover complex texts. We argue that a reliable graph structure should exhibit the following characteristics: the graph construction process should be stable, where *entities* serve as query entries, clearly defining the retrieval boundaries; *relations* act as guidance for further aligning

the query with the graph structure. Crucially, to completely preserve all relations, the relations should be inherent in the raw text rather than being extracted as concise expressions.

**Case 2: Cumulative semantic drift.** When propagating semantic embeddings with personalized PageRank (PPR) algorithms, the "similar but irrelevant" passages introduce interference, accumulating semantic bias during the retrieval process. For example, as shown in Figure 2b (right), in the "Rosalie Rauch" case, $p_2$ identifies Rosalie's father as Gustav Rauch, and $p_3$ notes that Gustav Rauch's son was born in Berlin. Since $p_2$ and $p_3$ are similar due to shared entities, PPR assigns an incorrect high importance to $p_3$ via "Gustav Rauch", resulting in $p_3$ being included in the context.

**Discussion 2.** Structure-augmented RAG relies on semantic embeddings for retrieval, where any semantic drift in these embeddings can potentially lead to retrieval failure. We summarize the main causes of failure into two key points: (1) Insufficient semantic expression: existing embedding models struggle to comprehensively capture the text details, resulting in inaccurate initial semantics; (2) Irrelevant information interference: explicit associations among knowledge are not always beneficial to queries, yet models struggle to distinguish them, causing unreliable semantic propagation. In summary, existing structure-augmented RAG leverages structures as a bridge between queries and passages. While it provides a fine-grained retrieval, it suffers from structural and semantic flaws. To address this, we propose KG-Translator, which frames retrieval as a translation task, translates queries into graph-level clues, enables better alignment between queries and passages, and without relying on complex graph construction or suboptimal embeddings.

## 3. Methodology

In the preliminaries, we analyze how structural and semantic flaws hinder the performance of structure-augmented RAG systems. Motivated by this, we propose a novel framework termed KG-Translator, which transforms the passage retrieval task into a clue translation task, as shown in Figure 3.

### 3.1. Overview

KG-Translator consists of three components: (1) Parsing-based KG Construction, which uses lightweight natural language processing (NLP) models to perform named entity recognition (NER) and syntactic parsing. It constructs hyperrelations by masking the entities from sentences, enabling the efficient construction of the directed hypergraph (ParseKG); (2) Structure-constrained KG Translation, which dynamically builds constraint indexes on the ParseKG to ensure the faithfulness of the translation process. In addition, we design a bidirectional translation mechanism to fully capture relational semantics, and employ structure-

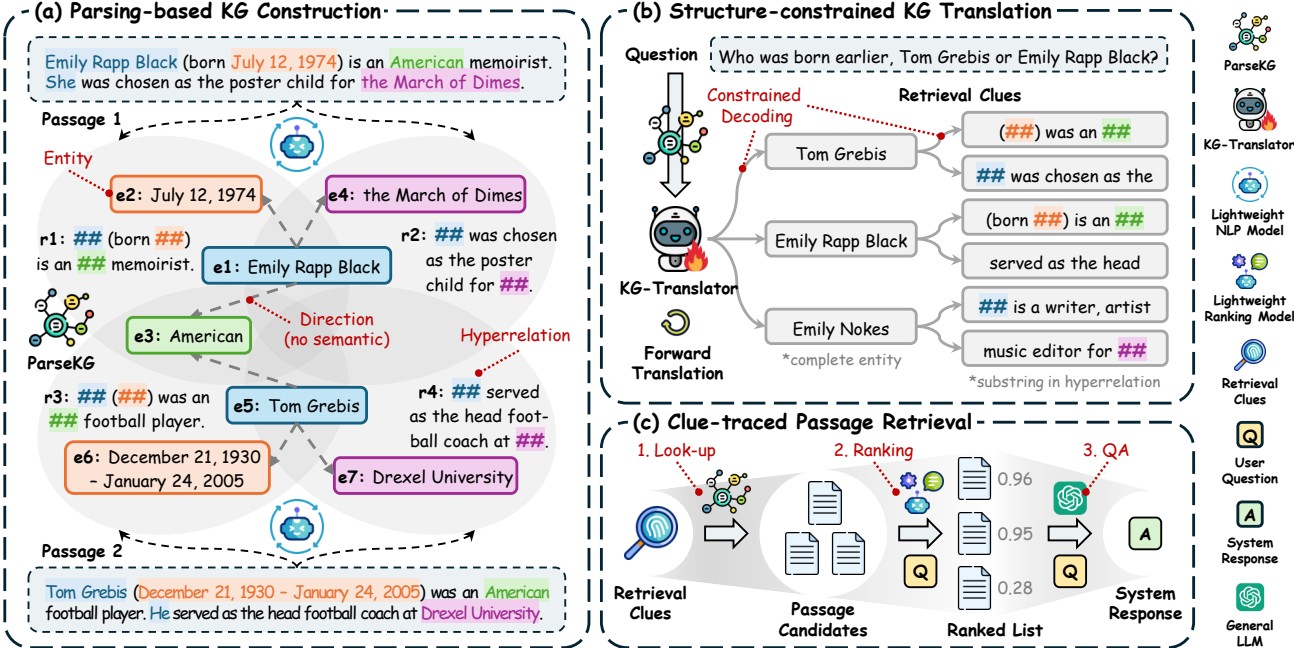

*Figure 3.* **The overall pipeline of our KG-Translator framework. (a) Parsing-based KG Construction:** We use a lightweight model to perform passage parsing, then construct hyperrelations from entity-masked sentences to build directed hypergraphs (ParseKG). **(b) Structure-constrained KG Translation:** We dynamically build constraint indexes on the ParseKG and constrain specialized LLM to ensure faithful clue generation. **(c) Clue-traced Passage Retrieval:** We trace these clues back to the original passages, then utilize a lightweight model for passage ranking to obtain contexts. Finally, we feed the contexts to a general LLM to derive the final answer.

aware fine-tuning to enhance the translation capability of LLMs; (3) Clue-traced Passage Retrieval, which traces the graph-level clues back to the original passages, and utilizes a lightweight model to rank the passages for precise retrieval.

### 3.2. Parsing-based KG Construction

Reliable knowledge graphs require the complete preservation of associations among knowledge and the provision of precise entries for queries. By utilizing lightweight NLP models (e.g., spaCy) to parse the corpora, we extract sentences and entities, identify the syntactic features of entities, and based on these results, construct a directed hypergraph named ParseKG. Specifically, given a passage set $\mathcal{P}$, we first perform the following three corpus parsing tasks:

**Sentence Segmentation.** We split each passage $p \in \mathcal{P}$ into sentences to obtain the sentence set $\mathcal{S} = \bigcup \mathcal{S}_p$ based on the punctuation (e.g., periods or exclamation marks).

**Named Entity Recognition (NER).** We extract all named entities from each sentence $s \in \mathcal{S}$ to obtain the entity set $\mathcal{E} = \bigcup \mathcal{E}_s$. Then, we derive the hyperrelation set $\mathcal{R} = \{r_s\}$, each $r_s$ being the text residue of $s$ after masking $\mathcal{E}_s$. For example, as shown in Figure 3, for the sentence "Emily Rapp Black (born July 12, 1974) is an American memoirist.", the text "## (born ##) is an ## memoirist." with $e_1$, $e_2$ and $e_3$ masked is defined as the hyperrelation, and "##" denotes the placeholder for the masked entities. Note

that $r$ and $s$ are in one-to-one correspondence, following the same notation convention (e.g., $\mathcal{E}_s$ and $\mathcal{E}_r$ are equivalent).

**Syntactic Parsing.** We parse the syntactic structure of each sentence $s \in \mathcal{S}$ and determines whether each entity $e_s \in \mathcal{E}_s$ serves as the syntactic subject. We define the subject entities as head entities $\mathcal{H} = \bigcup \mathcal{H}_s$ and non-subject entities as tail entities $\mathcal{T} = \bigcup \mathcal{T}_s$, where $\mathcal{H}_s \cup \mathcal{T}_s = \mathcal{E}_s$ and $\mathcal{H}_s \cap \mathcal{T}_s = \varnothing$. The design details and respective roles of the various parsing tasks are provided in Appendices F.2 and I.1, respectively.

**ParseKG Construction.** Based on the above parsing results, we define ParseKG as $\mathcal{G} = (\mathcal{E}, \mathcal{R}, \mathcal{D})$, where entities $\mathcal{E}$ are regarded as nodes and hyperrelations $\mathcal{R}$ are regarded as hyperedges. $\mathcal{D} = \bigcup \mathcal{D}_r$ denotes the direction set, with each $\mathcal{D}_r = (\mathcal{H}_r, \mathcal{T}_r)$ consisting of head entities $\mathcal{H}_r$ and tail entities $\mathcal{T}_r$, and the direction is denoted as $\mathcal{H}_r \to \mathcal{T}_r$. For the detailed statistical features of ParseKG, see Appendix E.

Unlike entity graphs or passage graphs constructed via OpenIE (Gutiérrez et al., 2025) or semantic similarity (Zhuang et al., 2026), ParseKG provides sufficient entry entities for queries to facilitate entity linking (De Cao et al., 2020), as well as retaining inter-entity correlations by taking entity-masked sentences as carriers and preserving complete relations in the text. Moreover, the construction of ParseKG solely relies on lightweight NLP models, which eliminates token overhead, enabling stable, efficient large-scale corpus processing and practical, feasible real-world applications.

## 3.3. Structure-constrained KG Translation

Although structure-augmented RAG enables retrieval via graph algorithms (e.g., PPR, GNNs) (Gutiérrez et al., 2025; Luo et al., 2025c), it still suffers from performance degradation caused by structural and semantic flaws (discussed in Section 2). To resolve this problem, we propose Structure-constrained KG Translation, which leverages constraint indexes including Trie (Cormen et al., 2022) and FM-index (Ferragina & Manzini, 2000) to ensure that LLMs translate queries into clues faithful to the ParseKG. Specifically,

Given a query $q$ and a ParseKG $\mathcal{G}$, our goal is to leverage LLMs for translating the query $q$ into structural clues $\mathcal{W} = \{(\hat{e}, \hat{r})\}$ on $\mathcal{G}$ to guide retrieval, where each clue $w \in \mathcal{W}$ consists of a translated entity $\hat{e}$ and a translated relation $\hat{r}$.

### 3.3.1. KG-CONSTRAINED INDEXING

To obtain valid clues, LLMs are required to follow ParseKG during decoding. Thus, we dynamically construct constraint indexes at test time to support the faithful translation.

**Entity-constrained Indexing.** The first stage of KG translation needs to determine the entry entity $\hat{e}$, which can be regarded as entity linking (De Cao et al., 2020). We construct a Trie (Cormen et al., 2022) for all candidate entry entities, where a Trie refers to a prefix tree that leverages prefix sharing for efficient string storage and retrieval, i.e.,

$$\mathcal{C}_{\mathcal{H}} = \text{Trie}(\text{tokenize}(\mathcal{H})), \tag{1}$$

where head entities serve as entries. We tokenize them with the tokenizer of LLM, and then store the tokens as Trie $\mathcal{C}_{\mathcal{H}}$.

**Relation-constrained Indexing.** Based on the translated entity and ParseKG, we retrieve the adjacent hyperrelations. The spans within these hyperrelations will be translated into relations. To support span decoding, we adopt the FM-index (Ferragina & Manzini, 2000), a compressed full-text index specialized in efficient substring retrieval. Similar to the construction of Trie, its process can be formulated as:

$$\mathcal{R}^+ = \mathcal{N}_+(\hat{e}) = \{r \mid \hat{e} \in \mathcal{H}_r\}, \tag{2}$$

$$\mathcal{F}_{\mathcal{R}^+} = \text{FM-index}(\text{tokenize}(\mathcal{R}^+)), \tag{3}$$

where $\mathcal{N}_+$ is the hyperrelation retrieval function that retrieves all out-hyperrelations connected to $\hat{e}$. After we tokenize the hyperrelations, we store them in FM-index $\mathcal{F}_{\mathcal{R}^+}$.

### 3.3.2. BIDIRECTIONAL KG TRANSLATION

Given the relational directionality, for example, "Bob is the father of Charlie" corresponds to the clue ("Bob", "is the father of"), and is also identifiable via the reversed form clue ("Charlie", "is the father of"). Though the latter misaligns in the relational direction, it still retains clues with

core semantics. Therefore, we capture comprehensive clues through the forward and reverse translations. Specifically,

**Forward Translation.** KG translation has two phases: entity translation and relation translation. In forward translation, entity translation leverages Trie $\mathcal{C}_{\mathcal{H}}$ to achieve complete entity decoding, and relation translation uses FM-index $\mathcal{F}_{\mathcal{R}^+}$ to implement hyperrelation span decoding. Formally,

$$P_\theta(w \mid q; \mathcal{G}) = \overbrace{P_e(\hat{e} \mid q; \mathcal{G})}^{\text{Entity Translation}} \cdot \overbrace{P_r(\hat{r} \mid q, \hat{e}; \mathcal{G})}^{\text{Relation Translation}} =$$
$$\prod_{i=1}^{|\hat{e}|} P_\theta(\hat{e}_i \mid q, \hat{e}_{1:i-1}) \cdot \mathcal{C}_{\mathcal{H}}(\hat{e}_i \mid \hat{e}_{1:i-1}) \cdot \tag{4}$$
$$\prod_{j=1}^{|\hat{r}|} P_\theta(\hat{r}_j \mid q, \hat{e}, \hat{r}_{1:j-1}) \cdot \mathcal{F}_{\mathcal{R}^+}(\hat{r}_j \mid \hat{r}_{1:j-1}),$$

where $P_\theta$ denotes the probability distribution of LLM parameterized by $\theta$, and $\hat{e}_i$ and $\hat{r}_j$ respectively denote the $i$-th token of the translated entity and the $j$-th token of the translated relation. $\mathcal{C}_{\mathcal{H}}$ and $\mathcal{F}_{\mathcal{R}^+}$ are constraint functions that respectively check whether the generated tokens satisfy the prefix constraint and the substring constraint. Formally,

$$\mathcal{C}_{\mathcal{H}}(\hat{e}_i \mid \hat{e}_{1:i-1}) = \mathbb{I}\{\exists \text{prefix}(\hat{e}_{1:i}, \hat{e})\}, \tag{5}$$
$$\mathcal{F}_{\mathcal{R}^+}(\hat{r}_j \mid \hat{r}_{1:j-1}) = \mathbb{I}\{\exists \text{substring}(\hat{r}_{1:j}, \hat{r})\}. \tag{6}$$

where $\mathbb{I}\{\cdot\}$ denotes the token-level indicator function.

Additionally, we also take the entities identified in the query $\{e_q\}$ as the translated entities. The number of clues is determined by beam size $(\alpha, \beta)$, where $\alpha$ and $\beta$ are the number of translated entities and translated relations, respectively.

**Reverse translation.** Considering that the clues derived from the reversed hyperrelations have a low occurrence frequency, we activate reverse translation when forward translation fails (details in Appendix J). Reverse translation retrieves the in-hyperrelations, i.e., in Equation 4, $\mathcal{R}^+$ is replaced with $\mathcal{R}^-$, where $\mathcal{R}^- = \mathcal{N}_-(\hat{e}) = \{r \mid \hat{e} \in \mathcal{T}_r\}$.

### 3.3.3. STRUCTURE-AWARE FINE-TUNING

To further enhance the KG translation capability of LLMs, we propose structure-aware fine-tuning. Given a query $q$, we optimize the LLM to autoregressively generate $\hat{e}$ and $\hat{r}$, such that the clue $w = (\hat{e}, \hat{r})$ can uniquely target the passages. The log-likelihood function is formulated as:

$$\mathcal{L} = \mathbb{E}_{(q,w) \sim \mathcal{O}} \log P_\theta(w \mid q)$$
$$= \mathbb{E}\left[ \log \prod_{i=1}^{|\hat{e}|} P_\theta(\hat{e}_i \mid q, \hat{e}_{1:i-1}) \cdot \prod_{j=1}^{|\hat{r}|} P_\theta(\hat{r}_j \mid q, \hat{e}, \hat{r}_{1:j-1}) \right], \tag{7}$$

where $\mathcal{O}$ denotes the training data, which consists of query $q$ and clue $w = (\hat{e}, \hat{r})$. We sample training data from PER (Liu et al., 2025), as it provides structured intermediate steps. We finally obtain 8,243 samples for training KG-Translator. For more details on the training data, refer to Appendix F.1.

*Table 1.* **QA performance (%).** We present the EM and F1 scores on four datasets. All RAG-based baselines employ all-MiniLM-L6-v2 for retrieval and GPT-4o-mini for graph construction and QA. We highlight the **best** and second-best results. We use teal (increase) and maroon (decrease) to represent the performance gaps (Δ) of each method relative to NaïveRAG, so as to demonstrate their structural advantages. For HotpotQA, 2WikiMQA, and MuSiQue, we evaluate a total of ten question types, with details reported in Appendix G.2.

| METHOD | HOTPOTQA | | 2WIKIMQA | | MUSIQUE | | SCALEQA | | AVG. | | Δ |
|---|---|---|---|---|---|---|---|---|---|---|---|
| | EM | F1 | EM | F1 | EM | F1 | EM | F1 | EM | F1 | |
| **VANILLA LLMS** | | | | | | | | | | | |
| LLAMA-3-8B (Grattafiori et al., 2024) | 36.90 | 44.10 | 28.80 | 36.18 | 4.12 | 12.14 | 9.13 | 17.32 | 19.74 | 27.43 | -20.18 |
| GPT-3.5-TRUBO (OpenAI, 2022) | 52.60 | 58.52 | 31.86 | 38.99 | 8.68 | 17.09 | 19.33 | 26.98 | 27.06 | 34.40 | -13.03 |
| GPT-4O-MINI (OpenAI, 2024) | 48.40 | 54.49 | 31.86 | 39.82 | 7.46 | 17.36 | 16.80 | 24.60 | 25.45 | 33.48 | -14.30 |
| **NAÏVE RAG** | | | | | | | | | | | |
| NAÏVERAG (TOP-1) | 52.20 | 58.26 | 33.74 | 40.59 | 11.20 | 17.94 | 17.87 | 24.91 | 27.85 | 34.60 | -12.54 |
| NAÏVERAG (TOP-3) | 63.70 | 70.00 | 44.38 | 51.74 | 19.50 | 27.74 | 26.60 | 34.37 | 37.64 | 45.12 | -2.38 |
| NAÏVERAG (TOP-5) | 67.90 | 72.00 | 45.55 | 52.90 | 21.55 | 30.29 | 30.13 | 38.24 | 40.11 | 47.41 | 0.00 |
| **STRUCTURE-AUGMENTED RAG** | | | | | | | | | | | |
| KGP (Wang et al., 2024) | 67.70 | 75.94 | 53.88 | 60.58 | 25.59 | 34.90 | 45.87 | 54.81 | 47.93 | 56.03 | +8.22 |
| RAPTOR (Sarthi et al., 2024) | 62.60 | 70.50 | 42.31 | 49.96 | 20.03 | 29.64 | 28.20 | 36.18 | 37.23 | 45.44 | -2.42 |
| GRAPHRAG (Edge et al., 2024) | 50.50 | 57.08 | 33.88 | 40.28 | 11.20 | 20.68 | 17.07 | 24.85 | 27.42 | 34.88 | -12.62 |
| LIGHTRAG (Guo et al., 2025) | 66.60 | 73.17 | 54.92 | 62.22 | 19.19 | 29.64 | 41.07 | 50.42 | 45.49 | 53.88 | +5.92 |
| HIPPORAG (Gutiérrez et al., 2024) | 69.00 | 76.80 | 65.93 | 72.54 | 19.42 | 28.86 | 46.27 | 55.73 | 51.20 | 59.35 | +11.51 |
| GFM-RAG (Luo et al., 2025c) | 69.80 | 77.22 | 62.40 | 69.36 | 27.19 | 35.45 | 55.33 | 64.04 | 54.08 | 61.84 | +14.19 |
| HIPPORAG2 (Gutiérrez et al., 2025) | 70.00 | 77.80 | 58.40 | 65.50 | 26.81 | 36.29 | 45.47 | 54.85 | 50.10 | 58.42 | +10.50 |
| PER-QA (Liu et al., 2025) | 69.80 | 77.47 | 63.86 | 69.90 | 31.07 | 40.27 | 48.13 | 57.18 | 53.64 | 61.41 | +13.76 |
| **KG-TRANSLATOR (OURS)** | | | | | | | | | | | |
| + LLAMA-3.2-1B | 72.10 | 80.26 | 72.99 | 78.52 | **33.89** | **43.32** | **59.67** | **68.33** | 60.83 | **68.45** | +20.88 |
| + QWEN-2.5-1.5B | **73.30** | **81.07** | **73.13** | **78.61** | 33.44 | 41.91 | 59.13 | 67.98 | **60.85** | 68.22 | +20.77 |

Since the entities in ParseKG provide entries for queries, and the hyperrelations constructed from entity-masked sentences preserve complete relational semantics, this provides support for the translation process of KG-Translator. Unlike the concise extract-based relation expressions, KG-Translator is capable of mining comprehensive relations via span-level decoding, avoiding structural and semantic flaws. Furthermore, the KG-constrained indexes ensure that the translation is faithful to ParseKG, while keeping it at constant time.

### 3.4. Clue-traced Passage Retrieval

KG-Translator translates queries into multiple clues on the ParseKG, and these clues provide precise guidance for the target passages. Thus, we trace them back to the original passages and sort the passages with a lightweight ranking model (e.g., all-MiniLM-L6-v2). Given a query $q$ and its corresponding clues $\mathcal{W}$, we can formalize this process as:

$$\hat{\mathcal{S}} = \{s \mid \hat{e} \in \mathcal{E}_s \wedge \hat{r} = r_s\}, \tag{8}$$

$$\hat{\mathcal{P}} = \bigcup_{s \in \hat{\mathcal{S}}} \{p \mid s \in \mathcal{S}_p\}, \tag{9}$$

$$\hat{\mathcal{P}}_k = \arg \text{Top-}k(\text{rank}(q, \hat{\mathcal{P}})), \tag{10}$$

where we select the Top-$k$ passages as supporting passages $\hat{\mathcal{P}}_k$ and set $k = 5$. Finally, we use the supporting passages as context and utilize general LLMs to generate answers.

Clue-traced passage retrieval benefits from the precise guidance provided by clues, which significantly narrows the retrieval scope and selects highly relevant passages. Thus, the lightweight ranking model is sufficient, where the similarity scores serve merely for ranking rather than retrieval.

## 4. Experiments

In this section, we conduct extensive experiments, aiming to answer the following four research questions: (1) **Q1 (Effectiveness)**: How does KG-Translator deliver comprehensive performance in QA tasks and on unseen datasets? (2) **Q2 (Efficiency)**: How efficient is KG-Translator in cost and time compared to existing structure-augmented RAG systems? (3) **Q3 (Robustness)**: How robust is KG-Translator in overcoming structural and semantic flaws? (4) **Q4 (Case Study)**: How does KG-Translator work in real-world scenarios? (Note that **Q4** is studied in Appendix H. To enable comprehensive evaluation of the translation paradigm, additional experiments are provided in Appendices G and I.)

### 4.1. Experiment Settings

**Datasets.** We evaluate the retrieval and QA capabilities of KG-Translator on four multi-hop QA datasets, including HotpotQA (Yang et al., 2018), 2WikiMQA (Ho et al., 2020), MuSiQue (Trivedi et al., 2022) from PER (Liu et al., 2025),

*Table 2.* **Ablation study**. We report F1 scores (%) on three datasets, ablating the structure-aware fine-tuning of KG-Translator and the reverse translation in the translation stage, while additionally considering a variant with reverse then forward translation order.

| Variant | HQA | 2Wiki | MSQ | Avg. |
|---|---|---|---|---|
| KG-Translator (Llama) | **80.26** | 78.52 | **43.32** | **68.50** |
| *w/o* structure-aware fine-tuning | 77.29 | 63.25 | 39.40 | 59.36 |
| *w/o* reverse translation | 77.38 | **78.65** | 37.12 | 66.08 |
| *w* reverse-then-forward | 72.41 | 68.36 | 36.32 | 59.79 |

*Table 3.* **Effectiveness on different-size backbones**. We report EM scores (%) of the Llama-3.2-Instruct (1B/3B) and Qwen-2.5-Instruct (1.5B/3B) series across three datasets. We use teal (increase) and maroon (decrease) to represent the performance gaps.

| Backbone | HotpotQA | 2WikiMQA | MuSiQue |
|---|---|---|---|
| Llama-1B | 72.10 | 72.99 | 33.89 |
| ↪ Llama-3B | **72.90** (+0.80) | **73.08** (+0.09) | **35.80** (+1.91) |
| Qwen-1.5B | **73.30** | **73.13** | 33.44 |
| ↪ Qwen-3B | 72.90 (-0.40) | 72.28 (-0.85) | **34.12** (+0.68) |

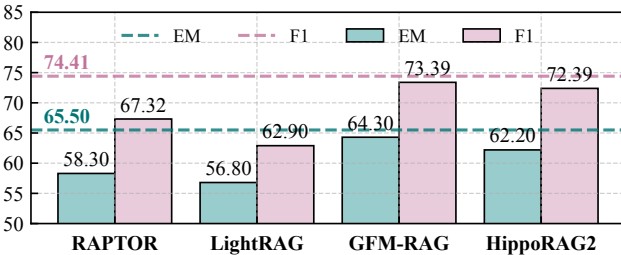

*Figure 4.* **Generalization experiment (%).** We transfer the KG-Translator to PopQA (Yang et al., 2018) for evaluating its generalizability for QA. The "horizontal line" represents KG-Translator.

and the self-constructed ScaleQA under a large-scale corpus setting. Following PER's settings, we test ten question types on HotpotQA (1,000), 2WikiMQA (2,125), and MuSiQue (1,313) to verify the cross-type adaptability. Numbers in parentheses are sample counts. To evaluate the generalizability of KG-Translator, we randomly select 1,000 samples for zero-shot transfer experiments on PopQA (Mallen et al., 2023). We follow HippoRAG (Gutiérrez et al., 2024), using supporting and distractor passages of the selected questions for corpus, excluding ScaleQA. We merge the corpora of three multi-hop datasets into one to serve as the corpus for ScaleQA. Details of datasets are provided in Appendix C.

**Baselines.** We compare KG-Translator with the 14 baselines grouped into three categories: (1) **Vanilla LLMs:** Llama-3.1-8B (Grattafiori et al., 2024), GPT-3.5-turbo (OpenAI, 2022), and GPT-4o-mini (OpenAI, 2024); (2) **Naïve RAG:** Select Top-1/3/5 supporting passages as contexts; (3) **Structure-Augmented RAG:** KGP (Wang et al., 2024), RAPTOR (Sarthi et al., 2024), GraphRAG (Edge et al., 2024), LightRAG (Guo et al., 2025), HippoRAG (Gutiérrez et al., 2024), GFM-RAG (Luo et al., 2025c), HippoRAG2 (Gutiérrez et al., 2025) and PER-QA (Liu et al., 2025). Baseline descriptions and configurations are available in Appendices D and L, respectively. Supplementary experiments on large embedding models are presented in Appendix G.5.

**Evaluation Metrics.** Following GFM-RAG (Luo et al., 2025c), we adopt Exact Match (EM) and F1 score as evaluation metrics for answer generation. For passage retrieval, we use Recall@2 (R@2) and Recall@5 (R@5) for evaluation. For answer extraction, we use unified prompts across all

methods. All prompts require the model to output concise answers in JSON format to ensure evaluation fairness.

**Implementation Details.** For KG-Translator, we adopt Llama-3.2-1B and Qwen-2.5-1.5B as backbones and employ the lightweight models en_core_web_trf (0.1B) and all-MiniLM-L6-v2 (22.7M) for graph construction and passage ranking, respectively. For all baselines, we use the same retriever (all-MiniLM-L6-v2) and generator (GPT-4o-mini) to ensure fairness. In addtion, for single-hop questions, we directly perform retrieval via KG-Translator without question decomposition; for multi-hop questions, we adopt an iterative decomposition strategy. Specifically, after each retrieval step, the LLM judges whether the acquired information is sufficient. If not, it generates the next sub-question to enable further retrieval. Details are listed in Appendix F.

### 4.2. Main Results (Q1)

**Obs. 1. KG-Translator exhibits superior performance compared to structure-augmented RAG.** As shown in Table 1, KG-Translator (Llama) outperforms the strongest baseline by 6.75% and 6.61% in average EM and F1 scores, respectively. Specifically, on 2WikiMQA, although HippoRAG2 builds on HippoRAG by enriching structures with passage nodes, it still suffers from structural flaws, leading to a 13.02% F1 score gap relative to KG-Translator (Llama). Despite well-designed structures, RAPTOR and GraphRAG are hindered by these structures, underperforming compared to NaïveRAG (Top-5). Additionally, retrieval performance and question-type-wise results in Appendices G.1 and G.2, respectively, further support this conclusion. This indicates that KG-Translator achieves precise retrieval via translation over ParseKG, without relying on expensive OpenIE.

**Obs. 2. Each module of KG-Translator is critical for optimal performance.** Table 2 shows that the full model outperforms its non-fine-tuned variant. This demonstrates that structure-aware fine-tuning effectively aligns queries with ParseKG, enabling the model to accurately translate clues. During translation, removing reverse translation results in a maximum performance drop of 6.2% (MuSiQue), while reversing the translation order leads to a more significant drop of 10.16% (2WikiMQA). This highlights that

*Table 4.* **Cost and efficiency per stage**. We report (1) time costs (s) and token costs ($\times 10^6$) for KG construction; (2) context number and performance for translation. Furthermore, we use arrows to highlight the improvements of KG-Translator over the second-best.

| METHOD | KG CONST. | | TRANSLATION | | |
|---|---|---|---|---|---|
| | TIME | TOKEN | CTX | R@5 | F1 |
| RAPTOR | 958.10 | 0.42 | 5.00 | – | 53.5 |
| LIGHTRAG | 3364.87 | 3.37 | – | – | 69.1 |
| HIPPORAG | 663.45 | 3.46 | 5.00 | 94.2 | 82.4 |
| GFM-RAG | 4788.57 | 3.28 | 5.00 | 89.7 | 78.0 |
| HIPPORAG2 | 667.68 | 3.46 | 5.00 | 88.6 | 76.4 |
| KG-TRANSLATOR | **288.56** | **0** | **3.74** | **96.1** | **87.0** |
| $\Delta$ (relative/absolute) | ↓56.5% | ↓100% | ↓1.26 | ↑1.9% | ↑4.6% |

relying solely on forward translation fails to capture complete knowledge, and reversing the order causes the model to deviate from query semantics, undermining the logic of structural matching. In addition, we provide detailed explanations and analyses in Appendix I on how ParseKG and KG-Translator collaborate within the translation paradigm.

**Obs. 3. KG-Translator exhibits strong zero-shot generalizability to unseen datasets.** We construct a ParseKG on PopQA and apply the KG-Translator to evaluate its zero-shot transferability. As shown in Figure 4, KG-Translator achieves improvements of 1.20% and 1.02% in terms of EM and F1 score, respectively. This demonstrates that the translation capability acquired by KG-Translator through structure-aware fine-tuning exhibits excellent zero-shot generalizability. Furthermore, KG-Translator can adapt to the knowledge structure of unseen dataset without additional training, indicating that the model captures general structural correlations between queries and clues, rather than relying on knowledge injection specific to a particular dataset.

**Obs. 4. KG-Translator follows the scaling law on more complex question types and corpus settings.** Compared with HotpotQA and 2WikiMQA, MuSiQue is more complex. For instance, it contains more 3/4-hop questions. As shown in Table 3, when scaling up both backbone models, KG-Translator consistently improves on MuSiQue. By contrast, on HotpotQA and 2WikiMQA, performance shows only marginal fluctuations with mixed slight gains and drops. Further analysis in Appendix G.3 reveals more noticeable improvements on harder cases such as 4-hop bridge-type questions, with performance gains of 5.12% (Llama) and 2.55% (Qwen). This indicates that the small-scale KG-Translator is sufficient for QA tasks in simple scenarios, and also exhibits optimization potential for complex scenarios.

### 4.3. Efficiency Analysis (Q2)

**Obs. 5. KG-Translator leverages a lightweight model for graph construction, cutting time significantly with zero token costs.** As shown in Table 4, the time and to-

*Table 5.* **Component-wise translation time (s)**. We measure the time cost of each component in the translation process on two-scale corpora (5K and 35K). Darker teal/maroon indicates higher time proportion per translation. Percentages are given in parentheses.

| COMPONENT | 5K-SCALE | 35K-SCALE |
|---|---|---|
| NER | 0.067 (5.2%) | 0.067 (2.8%) |
| ENT. INDEXING | 0.455 (35.3%) | 1.324 (56.0%) |
| ENTITY TRANSLATION | 0.173 (13.4%) | 0.173 (7.3%) |
| REL. INDEXING | 0.021 (1.6%) | 0.071 (3.0%) |
| RELATION TRANSLATION | 0.556 (43.2%) | 0.590 (24.9%) |
| RANKING | 0.016 (1.2%) | 0.141 (6.0%) |
| TOTAL TIME (PERCENT.) | 1.288 (100.0%) | 2.367 (100.0%) |

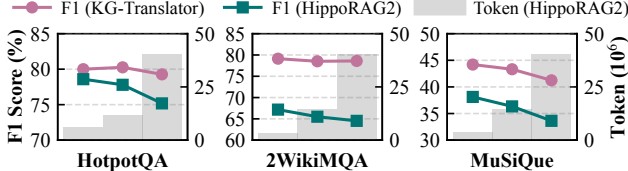

*Figure 5.* **Performance (%) scales with corpus expansion.** The curve denotes Small, Medium and Large scales in order. The graph construction process for KG-Translator incurs no token cost.

ken costs of ParseKG are only 288.56 seconds and 0 token, achieving a 56.5% and 100% reduction compared to the second-best method, respectively. Furthermore, structure-augmented RAG methods rely on OpenIE for graph construction, which inevitably leads to the knowledge loss. In contrast, ParseKG preserves complete knowledge through entities and hyperrelations (entity-masked sentences). Supported by KG-Translator, it ensures precise passage retrieval while maintaining a low-latency graph construction. This demonstrates that ParseKG is a practical knowledge graph.

**Obs. 6. KG-Translator can effectively identify passage relevance, enabling concise context without sacrificing performance.** Benefiting from constrained decoding on ParseKG, KG-Translator achieves constant-time clue generation. From the results in Table 4, KG-Translator obtains an average of 3.74 contexts per query, 1.26 fewer than other methods with a fixed number of contexts (typically set to 5). Meanwhile, the concise context is achieved without compromising performance. This indicates that KG-Translator can fully mine core clues aligned with the query, overcoming traditional retrieval's over-reliance on semantic similarity.

**Obs. 7. Scaling up the corpus brings no notable changes to the latency of most components within KG-Translator.** As shown in Table 5, (1) NER, entity translation, and relation translation show no significant runtime change, as their overhead depends on model inference time. (2) Relation-constrained indexing and ranking see a slight latency rise, as the beam size keeps them at a low magnitude. (3) The time cost of entity-constrained indexing is 2.9× higher and dominates total latency, as candidate entities grow substantially

*Table 6.* **Robustness of different embedding models**. We report R@5 (left) and F1 scores (right) on four scales of models, where all-MiniLM-L6-v2 is the default configuration of KG-Translator.

| MODEL | SIZE | HQA | 2WIKI | MSQ | AVG. |
|---|---|---|---|---|---|
| all-MiniLM-L6-v2 | 22.7M | 86/80 | 93/79 | 52/43 | 79/69 |
| all-mpnet-base-v2 | 0.1B | 87/80 | 93/79 | 52/42 | 80/68 |
| bge-large-en-v1.5 | 0.3B | 88/81 | 94/79 | 56/43 | 81/69 |
| gte-Qwen2-instruct | 1.5B | 86/80 | 93/78 | 49/41 | 78/68 |

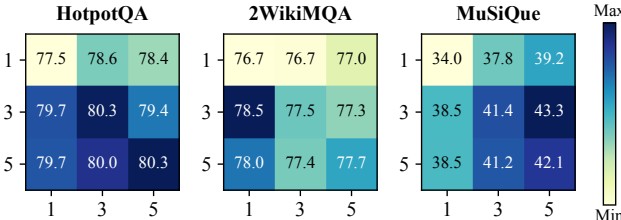

*Figure 6.* $\alpha, \beta$ **Sensitivity**. $(\alpha, \beta)$ determines the number of clues, where $\alpha$ **(y-axis)** defines the knowledge boundary associated with the query, and $\beta$ **(x-axis)** locks in the core semantics relevant to the query within this boundary. We conduct experiments and report F1 score on three datasets with $\alpha \in \{1, 3, 5\}$ and $\beta \in \{1, 3, 5\}$.

with corpus expansion. Notably, this latency can be mitigated by designing better caching mechanisms. Overall, a $7\times$ growth (5K→35K) in corpus size brings merely a $1.84\times$ increase (1.288s→2.367s) in total latency, demonstrating that KG-Translator can well support real-world applications.

**4.4. Robustness Analysis (Q3)**

**Obs. 8. With the continuous growth of the corpus size, KG-Translator maintains stable performance.** As shown in Figure 5, KG-Translator exhibits no significant performance degradation on HotpotQA and 2WikiMQA as the corpus scales up, whereas HippoRAG2 sees a drop of 3.40% and 2.64%, respectively. Although KG-Translator experiences a 2.97% drop on MuSiQue, this is still lower than HippoRAG2's 4.49%. Detailed results in Appendix G.4 also show that KG-Translator remains steady across 7 out of 10 question types, with fluctuations below 2%. This indicates that KG-Translator leverages translation to achieve better alignment between queries and graphs. By avoiding biases stemming from structural and semantic flaws, it gains more stable performance across corpora of varying scales.

**Obs. 9. KG-Translator accurately translates queries into clues, primarily relying on structure rather than embedding models.** From Table 6, all-MiniLM-L6-v2 (22.7M) exhibits a performance fluctuation of -1.1% to 0.2% across the three datasets when compared to larger models. This indicates that KG-Translator can maintain robust performance without excessively relying on embedding models. Even though the encoding capability of all-MiniLM-L6-v2 is far weaker than that of gte-Qwen2-instruct, KG-Translator ef-

fectively leverages structural advantages to translate clues, enabling the lightweight model to outperform larger models. Furthermore, we compare the performance of large embedding models and KG-Translator in Appendix G.5, demonstrating the outstanding performance of KG-Translator.

**Obs. 10. KG-Translator benefits from appropriate beam sizes and requires no extensive search to achieve performance gains.** From the results in Figure 6, for MuSiQue and 2WikiMQA, F1 scores increase by 3.6% and 0.8% respectively when $\alpha$ rises from 1 to 3 ($\beta = 3$); when $\beta$ rises from 1 to 3 ($\alpha = 3$), the former sees a 2.9% rise while the latter drops by 1.0%. Furthermore, though $(\alpha, \beta) = (5, 5)$ yields more clues, it delivers no significant gains relative to $(\alpha, \beta) = (3, 3)$. This indicates that the beam size needs to adapt to the knowledge distribution of corpus and that blind clue stacking introduces irrelevant context. In Appendix G.6, we further explain the roles of $\alpha$ and $\beta$ in translation.

## 5. Conclusion and Future Work

In this paper, we identify that existing structure-augmented RAG systems suffer from potential retrieval suspension and cumulative semantic drift. Motivated by this, we propose KG-Translator, a novel paradigm that translates queries into graph-level clues for precise passage retrieval. Specifically, KG-Translator first constructs a parsing-based knowledge graph (ParseKG) using a lightweight NLP model. On top of ParseKG, it adopts KG-constrained indexes and a specialized LLM to ensure that translated clues remain faithful to ParseKG. Finally, these translated clues are traced back to original passages, and a lightweight embedding model is employed to rerank contexts for answer generation. Extensive experiments and case studies demonstrate that our KG-Translator outperforms state-of-the-art baselines. In our future work, we will explore the following research direction: how to fully leverage the respective advantages of translation-based and matching-based retrieval to attain complementary gains for downstream tasks, thereby facilitating their broad adoption across diverse real-world scenarios.

## Acknowledgements

This work is supported by Xiamen Major Science and Technology Project (No. 3502Z20241028).

## Impact Statement

This work explores novel Retrieval-Augmented Generation (RAG) techniques to advance information retrieval capabilities in large language models (LLMs). Although our research has various social implications, we find no additional concerns deserving special focus beyond those generally associated with LLMs and information retrieval systems.

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

# Appendix

## A. Code and Dataset Availability

To ensure transparency and replicability, we have released our code, training and sample data at `https://github.com/GenIRAG/KG-Translator`. This repository contains the source code of KG-Translator, along with scripts for graph and indexing construction, model training, inference, and evaluation. Moreover, we provide a representative dataset sample and detailed instructions in our repository, enabling researchers to verify and replicate our proposed methodology.

## B. Related Work

### B.1. Structure-Augmented RAG System

Retrieval-Augmented Generation (RAG) alleviates hallucinations and factual inaccuracies in large language models (LLMs) (Huang et al., 2025; Zhang et al., 2025d) by introducing additional knowledge from external corpus (Borgeaud et al., 2022; Gao et al., 2023; Izacard et al., 2023). However, it still suffers from the omission of critical contextual information due to the fragmentation of disconnected document chunks (Han et al., 2024; Peng et al., 2025; Zhang et al., 2025b). To mitigate this, structure-augmented RAG has emerged as a novel paradigm, leveraging graph or tree structures to capture intricate dependencies within disparate knowledge sources or segments (Zhang et al., 2025b; Xiang et al., 2025; Xiao et al., 2025).

We categorize existing works into two main groups: (1) *Chunk-level methods*, which organize document segments into structured graphs or trees, and (2) *Entity-level methods*, which constructs fine-grained knowledge graphs via triple extraction.

(1) Chunk-level methods: These methods aim to preserve the global semantic context by organizing textual chunks into topological graphs (Munikoti et al., 2023; Li et al., 2025c) or hierarchical trees (Zhang et al., 2025a). Rather than treating chunks as isolated units, these approaches capture inter-passage relationships through contextual links (Wang et al., 2024) or recursive clustering (Sarthi et al., 2024), facilitating comprehensive information synthesis. However, such structural connections often remain coarse-grained and shallow, limiting models to identity the detailed information in the text.

(2) Entity-level methods: These methods construct explicit knowledge graphs (KGs) by extracting concise triples via open information extraction (OpenIE) (Edge et al., 2024; Guo et al., 2025; Zhao et al., 2025). Subsequently, they perform subgraph retrieval algorithms, such as personalized PageRank (PPR) and graph neural networks (GNNs), to locate specific factual evidence (Gutiérrez et al., 2024; 2025; Luo et al., 2025c). While offering fine-grained knowledge for passage retrieval, the graph construction remains unstable due to the over-reliance on the OpenIE of general LLMs (e.g., ChatGPT).

In addition, another category of research has leveraged the powerful planning capabilities of LLMs to structurally model queries. In this paper, we also regard it as a type of structure-augmented RAG and define it as Query-level methods. These methods shift focus towards decomposing complex questions into atomic sub-queries using structures such as reasoning trees (Xin et al., 2025; Shi et al., 2026) or directed acyclic graphs (DAGs) (Verma et al., 2025; Chen et al., 2026). By converting the questions into executable plans, these approaches provide structural and logical reasoning paths from a global perspective, facilitating fine-grained passage retrieval (Liu et al., 2025; Luo et al., 2025a). Despite advances, they still rely on embedding-based retrieval over unstructured corpora, failing to resolve the limitations of traditional RAG paradigms.

### B.2. Constrained Decoding for Retrieval

Traditional RAG systems mainly employ dense retrievers to implement embedding-based retrieval. Existing dense retrievers primarily utilize dual-encoder architectures to map queries and documents into a shared latent embedding space, retrieving relevant documents through similarity search (Karpukhin et al., 2020; Khattab & Zaharia, 2020). Given the excellent semantic understanding capabilities of LLMs, recent studies have attempted post-training on LLMs to enhance semantic representation (Li et al., 2023b; Muennighoff et al., 2024; Lee et al., 2025). Despite these advancements, these retrievers still face the embedding space bottleneck (Lee et al., 2022) and the limited query-document interaction (Wang et al., 2022).

The emergence of LLM constrained decoding presents a novel approach to solving these problems. As a promising strategy, constrained decoding has been widely adopted across various information retrieval tasks, including code search (Nadeem et al., 2022), conversational search (Li et al., 2023a; Liu et al., 2026), recommendation systems (Tan et al., 2024), and cross-modal retrieval (Li et al., 2024). Constrained decoding reformulates retrieval process as an autoregressive sequence generation task (Li et al., 2025b). Specially, constrained decoding is employed to prune the search space during inference (De Cao et al., 2020), ensuring that the results decoded by the large model exist in the corpus and can uniquely locate the document or passage. This process typically implemented via Trie for prefix constraints (Cormen et al., 2022) or FM-index for substring constraints (Ferragina & Manzini, 2000). More recently, researchers have integrated constrained decoding with

structure-augmented RAG. For example, based on Trie, GCR (Luo et al., 2025b) and DoG (Li et al., 2025a) conduct faithful reasoning constrained on knowledge subgraph, achieving more accurate KG question answering (KGQA) performance.

## C. Datasets

We evaluate KG-Translator on five datasets. For primary experiments, we include three multi-hop QA datasets, **HotpotQA** (Yang et al., 2018), **2WikiMultihopQA** (Ho et al., 2020), and **MuSiQue** (Trivedi et al., 2022). For generalization analysis, we incorporate a single-hop dataset, **PopQA** (Mallen et al., 2023). Following PER (Liu et al., 2025), we conduct evaluations separately on **10** question types within the multi-hop benchmarks to comprehensively demonstrate the performance. Furthermore, referring to the test data distribution of HippoRAG (Gutiérrez et al., 2024), we present an additional multi-hop QA dataset, named **ScaleQA**, built on a large-scale corpus to assess the performance during corpus expansion. Specifically,

- **HotpotQA** (Yang et al., 2018) is a large-scale benchmark to evaluate the multi-hop reasoning ability of models. It comprises 97k question-answer pairs and each is accompanied by 2 supporting passages along with several distracting passages, forcing models to filter irrelevant noise and perform cross-document inference to derive the correct answer.

- **2WikiMultihopQA** (Ho et al., 2020) is a comprehensive multi-hop QA dataset consisting of 192k instances. By leveraging structured templates from Wikidata and Wikipedia, this dataset provides fine-grained evidence and logical reasoning path for each question, effectively evaluating models' inference capability over complex reasoning chains.

- **MuSiQue** (Trivedi et al., 2022) is a challenging multi-hop QA benchmark containing 25k samples, where each instance is synthesized from 2 to 4 interconnected sub-questions to construct intricate reasoning chain. It demands that models execute coherent and systematic reasoning across multiple disparate passages while maintaining contextual consistency, thereby presenting rigorous multi-hop challenges and mitigating the reliance on cheatable shortcut-based reasoning.

- **PopQA** (Mallen et al., 2023) is an entity-centric QA dataset featuring 14k single-hop questions, generated by converting knowledge tuples from Wikidata via specialized templates. It is designed to investigate the ability of RAG systems to recall long-tail knowledge about low-popularity entities, which is often absent from LLMs' parametric memories.

- **ScaleQA** is a multi-hop QA benchmark designed to assess model performance in corpus extension scenarios. It focuses on 2-hop bridge QA, a task that graph-based RAG systems primarily target. The benchmark integrates the corpus of three source datasets to simulate real-world large-scale contexts, aligning with practical extension use cases.

*Table 7.* **Dataset (top) and graph (bottom) statistics.** We conduct evaluations on five representative benchmark datasets. $h \rightarrow r$ denotes the incident mapping from head entity to hyperrelation, and $r \rightarrow t$ denotes the incident mapping from hyperrelation to tail entity.

| | | **HOTPOTQA** | **2WIKIMQA** | **MUSIQUE** | **SCALEQA** | **POPQA** |
|---|---|---|---|---|---|---|
| **DATASET** | #OF DOMAIN | MULTI-HOP | MULTI-HOP | MULTI-HOP | MULTI-HOP | SINGLE-HOP |
| | #OF TYPES | 2 | 5 | 3 | 1 | 1 |
| | #OF QUERIES ($q$) | 1,000 | 2,125 | 1,313 | 1,500 | 1,000 |
| | #OF PASSAGES ($p$) | 9,633 | 12,406 | 12,718 | 34,733 | 9,743 |
| | #OF SENTENCES ($s$) | 38,625 | 44,270 | 46,815 | 129,649 | 56,809 |
| **GRAPH** | #OF TOTAL ENTITES ($e$) | 69,524 | 81,699 | 80,112 | 208,363 | 70,248 |
| | #OF TOTAL HYPERRELATIONS ($r$) | 38,625 | 44,270 | 46,815 | 129,649 | 56,809 |
| | #OF ENTRY ENTITIES ($h$) | 19,804 | 23,732 | 21,924 | 63,077 | 17,026 |
| | #OF NON-ENTRY ENTITIES ($t$) | 49,720 | 57,967 | 58,188 | 145,286 | 53,222 |
| | #OF $h \rightarrow r$ | 64,945 | 81,642 | 77,341 | 223,962 | 97,526 |
| | #OF $r \rightarrow t$ | 106,598 | 131,165 | 129,825 | 367,257 | 123,699 |

**ScaleQA Construction.** We analyze the public data of HippoRAG[2] and find that the proportions of bridge-type questions in HotpotQA, 2WikiMultihopQA, and MuSiQue are 81.1%, 75.6%, and 100.0%, respectively. Based on this statistical characteristic, we argue that bridge-type questions are the primary applicable task type for existing graph-based RAG systems (e.g., GFM-RAG, HippoRAG2, etc.). We thus sample 500 items each from the 2-hop bridge questions in the test sets of the three datasets, resulting in a total of 1,500 samples to construct ScaleQA. Furthermore, to verify the applicability of our method to real-world scenarios, we merge the corpus corresponding to the three datasets to achieve corpus extension.

---

[2]https://github.com/OSU-NLP-Group/HippoRAG

**Data Statistics.** As shown in Table 7, (1) For HotpotQA, we test 1,000 samples of two question-types, including: 2-hop bridge (Bri.[2H]) and 2-hop comparison (Comp.[2H]); (2) For 2WikiMultihopQA, we test 2,125 samples of five question-types, including: 2-hop bridge (Bri.[2H]), 2-hop comparison (Comp.[2H]), 2-hop inference (Inf.[2H]), 4-hop comparison (Comp.[4H]) and 4-hop bridge-comparison (B.C.[4H]); (3) For MuSiQue, we test 1,313 samples of three question-types, including: 2-hop bridge (Bri.[2H]), 3-hop bridge (Bri.[3H]), 4-hop bridge (Bri.[4H]). The statistics of test data for each type are shown in Table 8.

*Table 8.* **Detailed dataset (top) and graph (bottom) statistics of different question types.** We evaluate on various types of questions in HotpotQA, 2WikiMQA, and MuSiQue. The meanings of $h \rightarrow r$ and $r \rightarrow t$ are the same as in Table 7.

| | | HOTPOTQA | | 2WIKIMQA | | | | | MUSIQUE | | |
|---|---|---|---|---|---|---|---|---|---|---|---|
| | #OF TYPE | BRI.[2H] | COMP.[2H] | BRI.[2H] | COMP.[2H] | INF.[2H] | COMP.[4H] | B.C.[4H] | BRI.[2H] | BRI.[3H] | BRI.[4H] |
| DATASET | #OF QUERIES ($q$) | 500 | 500 | 500 | 500 | 500 | 125 | 500 | 500 | 500 | 313 |
| | #OF PASSAGES ($p$) | 4,973 | 4,687 | 2,570 | 3,567 | 2,735 | 8,69 | 3,199 | 7,706 | 4,010 | 2,153 |
| | #OF SENTENCES ($s$) | 18,521 | 15,997 | 10,433 | 10,613 | 11,841 | 2,845 | 10,287 | 27,721 | 15,531 | 8,103 |
| GRAPH | #OF TOTAL ENTITES ($e$) | 40,878 | 32,745 | 21,651 | 22,380 | 22,226 | 7,129 | 21,125 | 51,229 | 29,129 | 15,740 |
| | #OF TOTAL HYPERREL. ($r$) | 18,521 | 15,997 | 10,433 | 10,613 | 11,841 | 2,845 | 10,287 | 27,721 | 15,531 | 8,103 |
| | #OF ENTRY ENTITIES ($h$) | 10,917 | 9,208 | 5,542 | 6,421 | 5,909 | 1,632 | 5,798 | 13,783 | 7,413 | 3,977 |
| | #OF NON-ENTRY ENT. ($t$) | 29,961 | 23,537 | 16,109 | 15,959 | 16,317 | 5,497 | 15,327 | 37,446 | 21,716 | 11,763 |
| | #OF $h \rightarrow r$ | 32,975 | 26,070 | 20,163 | 16,912 | 29,106 | 4,479 | 15,324 | 45,481 | 25,848 | 13,573 |
| | #OF $r \rightarrow t$ | 59,231 | 46,929 | 30,862 | 28,362 | 37,077 | 8,522 | 31,807 | 76,149 | 43,446 | 23,126 |

## D. Baselines

In our experiments, we evaluate KG-Translator against 14 baselines, which can be categorized into three groups: Vanilla LLMs, Naïve RAG, and Structure-Augmented RAG. Below, we describe the details of each baseline:

(1) **Vanilla LLMs** rely solely on their parametric knowledge, serving as baselines to evaluate the models' zero-shot reasoning ability without access to external corpus. We include the following LLMs of varying families and sizes:

- **Llama-3.1-8B-Instruct** (Grattafiori et al., 2024) is a large-scale open-weight LLM from Meta, which is optimized for multilingual dialogue use cases, boasting robust instruction-following and strong contextual understanding capabilities.

- **GPT-3.5-trubo** (OpenAI, 2022) is a widely used closed-source model with robust performance across complex tasks.

- **GPT-4o-mini** (OpenAI, 2024) is a powerful flagship model of OpenAI that offers advanced reasoning capability and significantly reduced latency compared to GPT-3.5 series. We use gpt-4o-mini-2024-07-18 version.

(2) **Naïve RAG** follows the standard retrieve-and-read paradigm (Gao et al., 2023), which augments LLMs by retrieving multiple relevant passages from an unstructured corpus and concatenating them as external context for answer generation.

(3) **Structure-Augmented RAG** leverages graph or tree structures to explicitly construct the complex correlations and global contextual information across disparate knowledge. We include the following state-of-the-art baselines:

- **KGP** (Wang et al., 2024) creates a knowledge graph where nodes represent passages and edges capture semantic relations. It employs an LLM-based graph traversal agent as navigator to progressively gather relevant passages.

- **RAPTOR** (Sarthi et al., 2024) organizes text segments into a hierarchical tree via recursively embedding, clustering, and summarizing, allowing models to retrieve and integrate information across varying levels of abstraction.

- **GraphRAG** (Edge et al., 2024) builds an entity-level knowledge graph and generates hierarchical community summaries for related entity clusters. Both summaries and original documents are retrieved to provide global and local information.

- **LightRAG** (Guo et al., 2025) integrates graph structures with vector representations to achieve efficient retrieval. It also designs a dual-level retrieval paradigm to capture both fine-grained details and high-level semantic insights.

- **HippoRAG** (Gutiérrez et al., 2024) leverages OpenIE and embedding models to build a graph with relation and synonym edges, leveraging the personalized PageRank (PPR) algorithm to facilitate cross-passage retrieval.

- **GFM-RAG** (Luo et al., 2025c) follows the graph construction approach of HippoRAG but optimizes retrieval process by training a query-dependent Graph Foundation Model (GFM) to capture complex structural relationships.

- **HippoRAG2** (Gutiérrez et al., 2025) further extends HippoRAG framework by introducing passages nodes and context edges, linking queries to knowledge graph through triplets, as well as conducting LLM-based triplet filtering.

- **PER-QA** (Liu et al., 2025) adopts a Planner-Executor-Reasoner architecture, utilizing directed acyclic graphs (DAGs) to decompose questions into sub-queries and executing retrieval in topological order to collect supporting passages.

Additionally, we include three representative large embedding models as supplementary baselines to reflect the performance of traditional embedding-based RAG systems. These baselines provide a competitive point of comparison for assessing the effectiveness of our KG-Translator against strong semantic retrieval systems under standard settings.

(4) **Large Embedding Models** employ decoder-only architectures to map queries and documents into a shared latent embedding space, utilizing post-training on LLMs to enhance semantic representations for dense retrieval:

- **GTE-Qwen2-7B-Instruct** (Li et al., 2023b) encodes queries and passages into a shared semantic space using bidirectional attention, and generates high-quality text representations with strong capabilities in diverse retrieval tasks.

- **GritLM-7B** (Muennighoff et al., 2024) unifies generation and embedding via Generative Representational Instruction Tuning (GRIT), switching between causal (generation) and bidirectional (embedding) attention based on instructions.

- **NV-Embed-v2** (Lee et al., 2025) replaces standard mean pooling with a latent attention layer, removes causal masking during training for bidirectional understanding, and employs a two-stage contrastive instruction-tuning strategy.

## E. Graph Statistics

The statistics of ParseKGs constructed for each dataset are summarized in Tables 7 and 8. Specifically, we construct three graph variants of distinct scales to fully validate the method's robustness upon corpus scaling, including **small**-scale, **medium**-scale, and **large**-scale graphs. For small-scale graphs, refer to the statistics of each type in Table 8: their corpus scales range from 869 to 7,706, with approximately $5 \times 10^4$ entities and $2 \times 10^4$ hyperrelations. For medium-scale graphs, see the statistics for HotpotQA, 2WikiMQA and MuSiQue in Table 7: their corpus scales range from 9,633 to 12,718, with further expanded entity and hyperrelation counts of around $8 \times 10^4$ and $4 \times 10^4$, respectively. For large-scale graphs, refer to ScaleQA in Table 7, whose corpus scale is 34,733, with both entity and hyperrelation counts exceeding $1 \times 10^5$.

## F. Implementation Details

In this section, we introduce the training data construction, as well as the implementation details and core hyperparameter setting of ours KG-Translator and all baselines. For all used prompts in this paper, please refer to Appendix M.

*Table 9.* **Training and test set statistics.** When sampling the test set, we balance the samples across all question types to ensure comprehensiveness. We use the remaining samples (excluding test data) for training set construction. "split" denotes data partitioned from PER for training or testing, while "step" refers to the number of intermediate steps provided by PER. $^\dagger$ indicates data after deduplication.

| METHOD | HOTPOTQA | | 2WIKIMQA | | | | | MUSIQUE | | | TOTAL |
|---|---|---|---|---|---|---|---|---|---|---|---|
| | BRI.$^{2H}$ | COMP.$^{2H}$ | BRI.$^{2H}$ | COMP.$^{2H}$ | INF.$^{2H}$ | COMP.$^{4H}$ | B.C.$^{4H}$ | BRI.$^{2H}$ | BRI.$^{3H}$ | BRI.$^{4H}$ | |
| TOTAL | 1,166 | 1,052 | 1,036 | 1,013 | 1,011 | 125 | 1,019 | 1,018 | 641 | 313 | 8,394 |
| → TRAIN (SPLIT) | 666 | 552 | 536 | 513 | 511 | 0 | 519 | 518 | 0 | 0 | 3,815 |
| → TRAIN (STEP) | 2,242 | 1,954 | 1,127 | 1,063 | 1,034 | 0 | 2,080 | 1,236 | 0 | 0 | 9,865$^\dagger$ |
| → TEST (SPLIT) | 500 | 500 | 500 | 500 | 500 | 125 | 500 | 500 | 500 | 313 | 4,438 |

### F.1. Training Data Construction

We randomly sample data from HotpotQA, 2WikiMQA, and MuSiQue provided by PER (Liu et al., 2025) for testing, with the remaining data used as the training set (totaling 3,815 samples). PER provides intermediate steps for each multi-hop question, where each intermediate step can be treated as a single-hop question. Training and test set statistics are presented

in Table 9. We collect all single-hop questions (resulting in 9,865 samples after deduplication). Each single-hop question $q$ corresponds to a triplet composed of a head entity $h^*$, a supporting passage $p^*$, and a tail entity $t^*$. Specifically,

(1) **Entity Removal:** We use the lightweight model (en_core_web_trf) to identify all entities $\mathcal{E}_{p^*}$ in passage $p^*$, and remove all entities except $t^*$ to obtain $p^\star$; (2) **Relation Truncation:** We extract fixed-length spans from $p^\star$ using a sliding window (we set the window size to 10), requiring each span to contain the tail entity $t^*$, thus generating a candidate set $\{r^\star\}$; (3) **Similarity Filtering:** We calculate the similarity (based on all-MiniLM-L6-v2) between the single-hop question $q$ and each string $h^* \oplus r^\star$ ($\oplus$ represents string concatenation), select the $h^* \oplus r^\star$ with the highest similarity, and remove the tail entity $t^*$ from $r^\star$ to obtain the relation $r^*$. In the end, we obtain **8,243** clues $\mathcal{W} = \{(h^*, r^*)\}$ for model training.

### F.2. Implementation Details of KG-Translator

**Graph Construction.** We first leverage the spaCy model (en_core_web_trf) for corpus preprocessing, which encompasses sentence segmentation, named entity recognition, and syntactic parsing. The spaCy offers an integrated pipeline to facilitate the consistent and seamless implementation of these tasks. To ensure the rationality of entity decoding, we only allow entities whose entity types are not DATE, QUANTITY, PERCENT, TIME, LANGUAGE, MONEY, ORDINAL, NORP or CARDINAL and whose syntactic parsing results are nsubj (nominal subject) or nsubjpass (nominal passive subject) to serve as head entities. Then, we use a data structure declared based on NetworkX to store the constructed graph, and store token-level entities and hyper-relations through Trie (Cormen et al., 2022) and FM-index (Ferragina & Manzini, 2000) respectively.

**Model Training.** We implement KG-Translator based on LLaMA-Factory, and adopt Llama-3.2-1B-Instruct and Qwen-2.5-1.5B-Instruct as backbones for main experiments. We employ full-parameter fine-tuning on PER (details see Table 9), with a batch size of 64, a maximum sequence length of 2048, a learning rate of 1e-5, and 1 epoch. We use the cosine learning rate scheduler policy with a warmup ratio of 0.1. We conduct the training on a single Nvidia A800 80G GPU.

**Model Retrieval & QA.** The details of this part are as follows: (1) For translation, we use en_core_web_trf as the NER model to identify entities in the query, serving as a supplement to the translated entities. we set distinct beam sizes for different datasets to adapt to their respective corpora, with $(3, 3)$ for HotpotQA, $(3, 1)$ for 2WikiMQA, and $(3, 5)$ for both MuSiQue and PopQA. Subsets of ScaleQA adopt the configurations of their parent datasets. The maximum length of relation is 16 tokens. Referring to GFM-RAG (Luo et al., 2025c), we configure question decomposition with chain-of-thought (Wei et al., 2022) for multi-step QA to ensure the comprehensiveness of clues. (2) For passage ranking, we employ all-MiniLM-L6-v2 and select the Top-5 passages; all available passages are selected if fewer than 5 are retrieved. (3) For QA, we use gpt-4o-mini-2024-07-18 as the general LLM. All experiments are conducted on a single RTX 5090 32G GPU.

### F.3. Implementation Details of Baselines

We reproduce all baselines on the same retriever (all-MiniLM-L6-v2) and generator (gpt-4o-mini-2024-07-18) to ensure the fairness of experimental result comparisons. Notably, the open-source version of GFM-RAG is based on all-mpnet-v2, which differs from other baselines, and we adopt its multi-hop configuration. In addition, for retrieval, except for GraphRAG and LightRAG that are based on graph information summarization, we select the Top-5 retrieval results as the context for the other baselines; for QA, we design prompt to constrain the outputs in JSON format. See Appendix M for prompts. More detailed configurations of the baselines are provided in Appendix L, and we release their key hyperparameters.

## G. Additional Experiments

In this section, we provide additional experiments and analyses to comprehensively demonstrate the effectiveness and efficiency of our proposed KG-Translator. We report more detailed evaluation results of the main experiments in Appendix K.

### G.1. Retrieval Performance

To provide a more comprehensive assessment beyond QA performance, we demonstrate its corresponding retrieval performance. As shown in Table 10, we report the recall of the SOTA baseline and KG-Translator across four benchmarks.

**Obs. 11. KG-Translator demonstrates consistent retrieval superiority across diverse benchmarks.** As detailed in Table 10, KG-Translator surpasses the highest baseline by an average of 5.79% on R@2 and 5.14% on R@5. Notably, on ScaleQA, which features a large-scale corpus, KG-Translator still delivers an average gain of 7.00% compared to SOTA baseline. Such substantial improvements underscore that, by transforming embedding-based retrieval to clue translation,

*Table 10.* **Retrieval performance (%).** We use teal and maroon to represent the gaps (Δ) of each method relative to KG-Translator (Llama). For HotpotQA, 2WikiMQA, and MuSiQue, we evaluate a total of ten question types, with details reported in Appendix G.2.

| METHOD | HOTPOTQA | | 2WIKIMQA | | MUSIQUE | | SCALEQA | | AVG. | | Δ |
|---|---|---|---|---|---|---|---|---|---|---|---|
| | R@2 | R@5 | R@2 | R@5 | R@2 | R@5 | R@2 | R@5 | R@2 | R@5 | |
| **SOTA BASELINE** | | | | | | | | | | | |
| NAÏVE RAG (TOP-5) | 61.40 | 77.40 | 49.27 | 59.29 | 33.47 | 45.44 | 46.73 | 57.57 | 47.18 | 58.84 | -18.79 |
| HIPPORAG (Gutiérrez et al., 2024) | 70.60 | 85.90 | 69.66 | 86.69 | 37.47 | 47.42 | 55.37 | 70.07 | 59.09 | 73.68 | -5.42 |
| GFM-RAG (Luo et al., 2025c) | 41.70 | 78.65 | 51.49 | 84.67 | 21.78 | 43.15 | 42.53 | 71.90 | 41.01 | 71.25 | -15.67 |
| HIPPORAG2 (Gutiérrez et al., 2025) | 65.20 | 83.85 | 61.82 | 75.88 | 36.72 | 50.20 | 58.50 | 72.37 | 56.00 | 70.66 | -8.47 |
| PER-QA (Liu et al., 2025) | 61.50 | 75.75 | 61.92 | 79.45 | 38.49 | 51.97 | 52.73 | 66.63 | 54.35 | 69.51 | -9.87 |
| **KG-TRANSLATOR (OURS)** | | | | | | | | | | | |
| + LLAMA-3.2-1B | 74.35 | 86.30 | **76.25** | 92.46 | 36.65 | 51.88 | **66.77** | **78.10** | 64.78 | **78.82** | 0.00 |
| + QWEN-2.5-1.5B | **75.30** | **86.35** | 76.21 | **92.49** | **36.81** | **52.20** | 66.47 | 77.70 | **64.88** | 78.81 | +0.05 |

KG-Translator achieves precise passage retrieval, significantly boosting QA with reliable and comprehensive contexts.

## G.2. Effectiveness on Different Question Types

We conduct a detailed analysis of KG-Translator across various 10 question types to evaluate its robustness. As shown in Figure 7a, we report the QA performance of ours KG-Translator compared with SOTA baselines. The complete experimental data is reported in Tables 18 and 19.

**Obs. 12. KG-Translator exhibits broad applicability across diverse question types.** As shown in Figure 7a, even against the top-performance across all baselines, KG-Translator still demonstrates exceptional competitiveness, achieving SOTA performance across 8 question types on three datasets with a maximum performance improvement of 8.1% (2Wiki Inf.[2H]). Although its performance is not optimal on certain types, it still maintains competitive performance, with the gaps being merely 0.6% (HQA Bri.[2H]) and 2.7% (2Wiki Comp.[2H]) respectively. This demonstrates that KG-Translator exhibits strong cross-type adaptability and robustness. We attribute this to the translation process achieving precise alignment between queries and the ParseKG: the translation process does not overfit the distribution of any specific type of data, but instead deeply captures the associative relationships between the semantic features of queries and the structural features of graph-level clues.

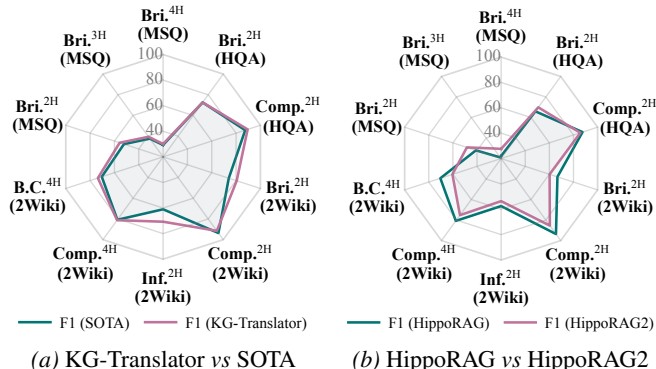

*(a)* KG-Translator *vs* SOTA     *(b)* HippoRAG *vs* HippoRAG2

*Figure 7.* **F1 scores (%) across different question types.**

**Obs. 13. Different types exhibit distinct characteristics, with overfocusing on one type impairing performance on the others.** For instance, the bridge-type requires precise identification of intermediate entities, whereas the comparison-type places greater emphasis on identifying and comparing the attributes of multiple entities (Ho et al., 2020). As shown in Figure 7b, we observe performance degradation in the HippoRAG series of methods: HippoRAG2 achieved improvements of 3.9% and 7.5% over HippoRAG on 2-hop bridge-type for HotpotQA and MuSiQue, respectively, but suffered declines of 1.8% and 8.2% on 2-hop comparison-type for HotpotQA and 2WikiMQA, respectively. We analyze the test data distribution of HippoRAG and found that the proportion of bridge type is indeed higher, accounting for 81.1% and 75.6% of HotpotQA and 2WikiMQA, respectively. This indicates that although HippoRAG2 introduces passage nodes to enrich the graph structure for better identification of inter-entity associations, inaccurate graph construction significantly impairs its performance on non-bridge questions, rendering it unable to adapt to different question types. This indirectly validates the effectiveness of KG-Translator in its architectural design, as it is able to capture the common patterns across different types.

## G.3. Effectiveness on Different Scale Models

To investigate the impact of backbone size on KG-Translator, we analyze its scaling behavior across the Llama (1B, 3B) and Qwen (1.5B, 3B) series in Table 3. We find that increasing model size only brings marginal gains or even causes performance

declines on some datasets. We provide detailed results in Table 11 to further clarify the scaling law of KG-Translator.

**Obs. 14. Scaling up the backbone size yields only marginal gains for KG-Translator.** As shown in Table 3, scaling the Llama backbone from 1B to 3B yields moderate performance improvements. For instance, on MuSiQue, Llama-3B achieves an EM of 35.80%, improving by 1.91% over Llama-1B, while on HotpotQA, the gain is marginal. This trend indicates that 1B models already capture the essential capability required for mapping queries into graph-level clues. The observed scaling behavior aligns with findings in constrained settings, where increasing model capacity often yields subproportional improvements rather than the proportional gains typically seen in open-ended generation tasks (Urbizu et al., 2023). It is worth noting that the Llama series represents a larger relative increase in size (3×) compared to the Qwen series, which only doubles in size. We observe that the Qwem-3B model does not consistently outperform its 1.5B version. For example, it exhibits a performance drop on HotpotQA and 2WikiMQA. This aligns with prior findings that performance gains are not always consistent as model size increases, and can vary significantly based on specific task characteristics (Ivgi et al., 2022). Based on these results, we argue that marginal gains do not justify the notable increases in inference latency and computational overhead of larger backbones (e.g., 3B-scale, 7B-scale). Thus, the model with 1B parameters is the optimal efficiency sweet spot for the KG-Translator framework, balancing translation accuracy and computational resource costs.

**Obs. 15. Under complex question types and corpus settings, scaling up KG-Translator's backbone achieves notable performance gains.** As shown in Table 11, we observe significant performance gains from model scaling on the 4-hop-bridge questions of MuSiQue: 5.12% for Llama and 2.55% for Qwen. We analyze this phenomenon in detail: (1) Question type: Compared with 2-hop questions, 4-hop questions require KG-Translator to have stronger translation capabilities to filter out irrelevant clues. (2) Corpus setting: Compared with HotpotQA, MuSiQue contains more passages on the same topic. For instance, there is only one passage about "Einstein" in HotpotQA but five in MuSiQue. Therefore, KG-Translator still follows the scaling law under model scaling on more complex question types and corpora, indicating its performance is not fully limited by ParseKG.

*Table 11.* **Effectiveness of different-scale models under various question types and corpus settings**. We report EM scores across Llama and Qwen series.

| **BACKBONE** | **HQA** | **MSQ** | | |
|---|---|---|---|---|
| | **BRI.**[2H] | **BRI.**[2H] | **BRI.**[3H] | **BRI.**[4H] |
| LLAMA-1B | 59.00 | 46.00 | 28.80 | 22.68 |
| LLAMA-3B | 60.20 | 47.60 | 29.00 | 27.80 |
| Δ (1B → 3B) | +1.20 | +1.60 | +0.20 | +5.12 |
| QWEN-1.5B | 60.20 | 47.20 | 27.00 | 21.73 |
| QWEN-3B | 60.80 | 45.40 | 29.00 | 24.28 |
| Δ (1.5B → 3B) | +0.60 | -1.80 | +2.00 | +2.55 |

### G.4. Effectiveness on Different Scale Corpora

To further validate the robustness of our proposed KG-Translator against scaling corpus sizes, we present a comprehensive evaluation across ten question types and three scales (approximately 5k → 10k → 35k) on three benchmarks, as shown in Figure 8. Detailed graph statistics are shown in Tables 7 and 8.

**Obs. 16. Our KG-Translator maintains superior stability over structure-augmented RAG as corpus scale increases.** As illustrated in Figure 8, KG-Translator demonstrates significantly resilient performance trajectories across all question types regardless of the incremental corpus scales. Specifically, as corpus expands from small to large, HippoRAG2 suffers from performance degradation on 2WikiMQA Bri.[2H], with F1 score drop by up to 9.5% . In contrast,

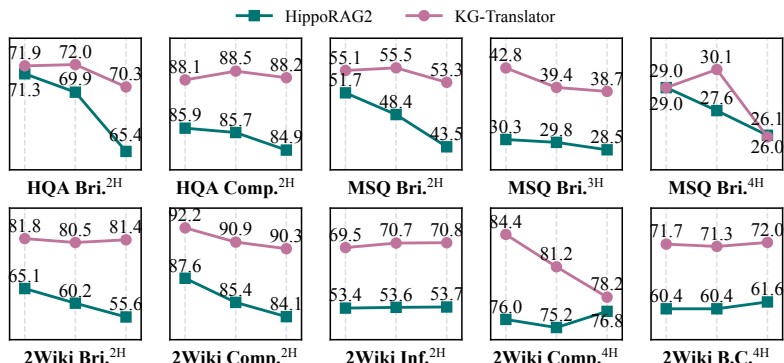

*Figure 8.* **F1 scores (%) across different question types as the corpus expands.** Each line represents scales from small (~5k) → medium (~10k) → large (~35k).

KG-Translator remains steady on 7 out of 10 question types, exhibiting fluctuations of less than 2%. Although it exhibits a performance drop of 4.1%, 2.9%, and 6.2% on MuSiQue Bri.[3H] & Bri.[4H] and 2WikiMQA Comp.[4H] respectively, its performance on large-scale corpora still outperforms HippoRAG2. We attribute this drop primarily to error accumulation: these question types involve more retrieval hops, and inaccurate first-hop retrieval can easily compromise subsequent retrieval. In brief, benefiting from a reliable ParseKG and precise KG translation, our framework effectively mitigates the structural flaws in the graph introduced by corpus expansion, achieving exceptional robustness across varying scales.

## G.5. Performance Comparison with Large Embedding Model

To further assess the advantages of KG-Translator over expressive embedding models, we incorporate representative large embedding models as the retriever for NaïveRAG (Top-5) to establish more competitive baselines. Specifically, we choose the representative GTE-Qwen2-7B-Instruct (Li et al., 2023b), GritLM-7B (Muennighoff et al., 2024) and NV-Embed-v2 (Lee et al., 2025) to evaluate QA and retrieval performance on ScaleQA, with the results shown in Table 12.

**Obs. 17. KG-Translator outperforms large embedding models in challenging large-scale corpus settings.** On the challenging ScaleQA benchmark coupled with an extensive corpus, KG-Translator consistently transcends all baselines across all evaluation metrics. Specifically, KG-Translator achieves remarkable performance gains in QA tasks, with a maximum improvement of 14.54% on EM and 12.85% on F1. In terms of retrieval performance, KG-Translator surpasses the strongest large-scale retrievers by up to 4.07% on R@2 and 2.40% on R@5. Notably, KG-Translator attains these results while only utilizing LLMs with much smaller scales of 1B (Llama)

*Table 12.* **QA and retrieval performance comparison with Large Embedding Model.** We report four evaluation metrics on ScaleQA.

| METHOD | QA | | RETRIEVAL | |
|---|---|---|---|---|
| | EM | F1 | R@2 | R@5 |
| GTE-QWEN2-7B-INSTRUCT | 42.13 | 52.72 | 59.57 | 72.83 |
| GRITLM-7B | 45.13 | 55.48 | 62.70 | 75.70 |
| NV-EMBED-v2 | 44.53 | 54.50 | 61.93 | 75.40 |
| KG-TRANSLATOR (LLAMA) | **59.67** | **68.33** | **66.77** | **78.10** |
| KG-TRANSLATOR (QWEN) | 59.13 | 67.98 | 66.47 | 77.70 |

to 1.5B (Qwen), maintaining a significantly lighter computational footprint compared to large embedding models with 7B parameters. Besides, we find that the performance gap in QA is more pronounced than that in retrieval. We attribute this to the targeted translation of KG-Translator, which enables concise supporting passages corresponding to clues and thus mitigates the impact of knowledge conflicts in the context on the model's reasoning process during QA. Overall, these experimental results underscore that despite the adoption of larger, more expressive embedding models, the traditional RAG paradigm remains constrained by the inherent flaws of embedding-based semantic matching. Conversely, by leveraging structural query-to-clue translation, our proposed KG-Translator achieves precise passage retrieval within vast search spaces, demonstrating its outstanding superiority in both QA accuracy and retrieval quality.

**Obs. 18. Structure-augmented RAG with large embedding models still underperforms KG-Translator using a small one.** We conduct experiments on HippoRAG equipped with large embedding models (LEMs). As shown in Table 13, we find that even with stronger embeddings, HippoRAG still underperforms KG-Translator by 7.67% on average. Furthermore, HippoRAG shows no obvious performance gap when adopting small or large embedding models. One possible reason is that: In typical RAG systems, LEMs serve as retrievers and are fine-tuned for passage-level retrieval tasks, endowing them with powerful semantic comprehension capabilities. In structure-augmented RAG systems, methods such as HippoRAG adopt embedding models to implement entity linking. Nevertheless, LEMs fail to

*Table 13.* **QA performance comparison with HippoRAG using different embedding models.** We report F1 scores across three multi-hop QA datasets and the average performance.

| METHOD | HQA | 2WIKI | MSQ | AVG. |
|---|---|---|---|---|
| HIPPORAG (Gutiérrez et al., 2024) | | | | |
| + all-MiniLM-L6-v2 | 76.80 | 72.54 | 28.86 | 60.58 |
| + GTE-Qwen2-7B | 75.88 | 69.28 | 26.89 | 58.23 |
| + NV-Embed-v2 | 77.48 | 72.40 | 29.41 | 60.83 |
| KG-TRANSLATOR (OURS) | | | | |
| + all-MiniLM-L6-v2 | **80.26** | **78.52** | **43.32** | **68.50** |

fully exploit their superior semantic capabilities when deployed for entity linking, since standalone entities lack adequate contextual information. For example, it is difficult to disambiguate the entity "apple" without supplementary context like "fruit" or "smartphone brand". This trend has also been confirmed in prior studies such as GFM-RAG (Luo et al., 2025c).

## G.6. Beam Size Sensitivity

We report the retrieval sensitivity of different beam sizes on three datasets in Figure 9, which is different from the QA analysis in Figure 6. More comprehensive experimental data on each type can be found in Tables 24 and 25.

**Obs. 19. $\alpha$ controls the number of translated entities, effectively defining the breadth of the knowledge for retrieval.** Across all datasets, increasing $\alpha$ improves R@5 initially before gradually saturating. As observed in HotpotQA with $\beta = 1$, increasing $\alpha$ from 1 to 3 leads to a significant performance improvement, with the average R@5 rising from 73.1% to 83.9%. A similar trend is observed on 2WikiMQA, where R@5 rises from 85.2% to 92.5% as $\alpha$ increases from 1 to 3. This notable gain indicates that a single translated entity is often insufficient to cover the necessary clues. Expanding the number of entry entities establishes complementary coverage. When $\alpha$ is further increased to 5, performance gains stabilize, suggesting

that the Top-3 translated entities have already captured the core knowledge boundary, essentially saturating the recall potential of entity-based anchoring. The value of $\alpha$ lies in effectively expanding the retrieval boundary by introducing multiple entry points, which is particularly critical for questions involving multiple entities.

**Obs. 20. $\beta$ controls the number of translated relations, affecting semantic relevance within the translated-entity-centered space.** Unlike $\alpha$, $\beta$ does not expand the knowledge boundary but focuses retrieval directions

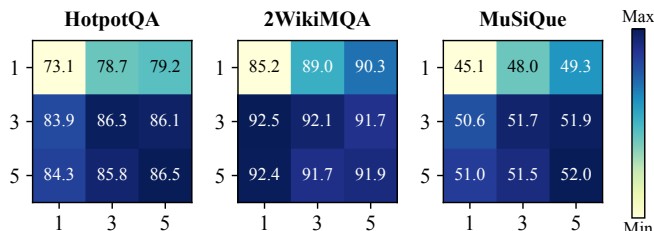

*Figure 9.* $\alpha, \beta$ **Sensitivity of retrieval**. The y-axis and x-axis represent $\alpha$ and $\beta$ respectively. We report R@5 (%) on three datasets.

with abstract-level semantics. For a narrow knowledge boundary (e.g., $\alpha = 1$), increasing $\beta$ from 1 to 3 leads to a steady rise in R@5 by 5.6%, 3.8%, and 2.9% on the three datasets, respectively. This trend demonstrates that translating multiple relations helps the model better capture semantic interactions between queries and passages, and filter out passages that merely mention the same entities without contributing to answering the question. For a wide knowledge boundary (e.g., $\alpha = 3$), increasing $\beta$ results in an initial performance increase that then stabilizes on HotpotQA, while no significant improvement is observed on the other two datasets. This may be attributed to the relations giving rise to similar semantics (e.g., "is a" and "is an"), which makes it difficult for the model to further distinguish between subtle semantic differences.

### G.7. Error Analysis

We analyze qualitative error cases of our KG-Translator framework to identify potential limitations, as shown in Table 14. These errors can be categorized into two types: (1) Ordinary noun omission, and (2) Excessive similar relations.

*Table 14.* **Two error types of KG-Translator. Case 1** (top): Ordinary noun omission; **Case 2** (bottom): Excessive similar relations. Olive, teal and maroon denote the core entities in the query, translated entities and the core semantics of translated relations, respectively.

| CASE 1: ORDINARY NOUN OMISSION | |
|---|---|
| QUESTION | Which **car brand** was founded on November 3, 1911? |
| ANSWER | Chevrolet |
| CLUE | ("**Ford Motor Company**", "## four-wheel drive **is the brand name of** a selectable automatic") 
 ("**Ford Genk**", "The plant **became the only car manufacturing plant in** ## after ## closed at ##") |
| CONTEXT | **1. Volvo Car Gent:** ... only car manufacturing plant in Flanders ... closed at the end of 2014. (Score: 0.86) 
 **2. ControlTrac:** ... is the brand name of a ... drive system offered by Ford Motor Company. (Score: 0.65) |
| PREDICTION | Volvo (✗) |

| CASE 2: EXCESSIVE SIMILAR RELATIONS | |
|---|---|
| QUESTION | Who received **the Bharat Ratna** before becoming President of India? |
| ANSWER | A. P. J. Abdul Kalam |
| CLUE | ("**the Bharat Ratna**", "## **was awarded several high awards** during his life ...") 
 ("**the Bharat Ratna**", "## – 12 April 1962) **was an ## chief civil engineer**.") 
 ("**the Bharat Ratna**", "In ##, the ## **government honoured him** with ##'s highest civilian award") 
 ... 
 ("**Bharat Ratna**", "## **Aerospace scientist Professor Author Awards** ## (##) ## Medal (##)") |
| CONTEXT | **1. Bharat Ratna:** The Bharat Ratna (Jewel of India) is the highest civilian award of the Republic ... (Score: 0.70) 
 **2. Bharat Ratna:** There is no formal provision that recipients of the Bharat Ratna should be ... (Score: 0.67) 
 **3. Bharat Ratna:** The first recipients of the Bharat Ratna were politician C. Rajagopalachari ... (Score: 0.64) 
 ... 
 *6. A. P. J. Abdul Kalam: ... Aerospace scientist Professor Author Awards Bharat Ratna (1997) ... (Score: 0.48)* |
| PREDICTION | C. Rajagopalachari (✗) |

**Obs. 21. Only ordinary nouns in queries can lead to retrieval failure.** To ensure the accuracy of entries, we only consider named entities during graph construction, which typically define a clear retrieval boundary. However, a query may contain only ordinary nouns. For instance, in Case 1, the question requires identifying "the car brand founded on November 3, 1911".

Although KG-Translator translates entities "Ford Motor Company" and "Ford Genk" relevant to car brands, the fixed beam size prevents it from enumerating all car brands. The inaccurate entry thus renders relation translation meaningless, even though it accurately captures query-relevant semantics (e.g., "is the brand name of"), thereby leading to retrieval failure.

**Obs. 22. Broad retrieval boundaries can lead to QA failure.** Translated entities may be overly prevalent in the graph, leading them to be distributed across a vast number of hyperrelations. The spans of these hyper-relations can be highly semantically similar, which makes it difficult for KG-Translator to distinguish which clues are useful for the query. For example, in Case 2, for the query "Who received the Bharat Ratna before becoming President of India?", KG-Translator accurately identifies the entry entity ("The Bharat Ratna"). However, this entity is linked to numerous hyperrelations, a large number of which contain semantics related to "awards". While KG-Translator can successfully translate relations semantically close to the query (e.g., "was awarded several high awards", "Aerospace scientist Professor Author Awards"), the excessive similar hyperrelations result in a massive number of passages being retrieved after clue tracing. After ranking, only the Top-5 passages are selected as the context for QA, yet the passage that supports answering the query is ranked 6th, ultimately leading to erroneous QA outcomes. Notably, this phenomenon validates our rationale for excluding ordinary nouns during graph construction: when an translated entity is associated with an excessive number of relations whose core semantics are similar, the correct signal is easily lost due to the abundance of distractors, leading to QA failure.

## H. Case Study

To provide an intuitive understanding of the working mechanism of KG-Translator, we conduct a detailed qualitative analysis on two representative running examples, as shown in Table 15. We present the following observations:

**Obs. 23. KG-Translator can simultaneously translate multiple topics mentioned in a query to obtain comprehensive clues.** Case 1 presents a running example of single-step translation for the question: "Are Ramesht and Salajwe both located in the same country?". This question involves two topics: the residence of Ramesht and that of Salajwe. KG-Translator acquires precise clues related to both Ramesht and Salajwe simultaneously in one single translation, and these clues encapsulate the core semantics of the intent ("place of residence"), i.e., "## is a village in ##". These clues are then traced back to the original passages for ranking, with the final context retrieved therefrom; the top two context are the supporting passages. Furthermore, although an irrelevant clue related to "Rahim Aga Khan" is translated (as the number of clues is constrained by the beam size), this clue is ranked last in the final context due to its low semantic relevance to the query.

**Obs. 24. KG-Translator disambiguates the query via end-to-end translation to accurately capture the query intent.** For the question in Case 2: "What football club plays in the area between the old toll gates: Brook Bar and Trafford Bar?", its topics "Brook Bar" and "Trafford Bar" are not directly modeled in ParseKG, and thus the translated clues do not treat them as entries directly. However, as KG-Translator leverages the strong semantic comprehension capability of LLMs to enable end-to-end translation, the two topics are disambiguated during the translation process, yielding clues related to "Old Trafford, Greater Manchester". These clues also encapsulate the core semantics of the query intent ("area"): "two old toll gates" and "is an area of". After clarifying the first intent, we use a LLM to perform question decomposition, resulting in the follow-up query: "What football club is based in Old Trafford, Greater Manchester?", which focuses on the second intent ("football club"). It can be observed that KG-Translator leverages its robust translation capability once again to translate clues that do not explicitly appear in ParseKG, and these clues incorporate the core semantics of the query: "football club". Finally, KG-Translator gathers sufficient supporting evidence and leverages a general LLM to achieve accurate QA.

## I. From the NLP Perspective: Experiments and Discussions on KG-Translator

In this section, we conduct an insightful analysis of KG-Translator from the NLP perspective. Specifically, (1) **The Role of Lightweight NLP Model:** We analyze the roles of various corpus parsing tasks. (2) **Entity Linking:** We investigate the entity translation of KG-Translator. (3) **Semantics:** We compare the representational capabilities of traditional semantic embeddings and clues. (4) **From Clues to Triples:** We modify the translation process of KG-Translator to enable it to generate triple clues instead of binary clues, so as to explore the intrinsic nature of its translation in greater depth.

### I.1. The Role of Lightweight NLP Model

We analyze core corpus parsing tasks or results to demonstrate how ParseKG facilitates KG-Translator's translation.

**The Role of Sentence Segmentation.** When constructing ParseKG, hyperrelations are defined at the sentence-level rather

*Table 15.* **Case study of KG-Translator. Case 1** (top): Single-step translation; **Case 2** (bottom): Multi-step translation. Olive, teal and maroon denote the core entities in the query, translated entities in clues, and the core semantics of relation, respectively.

| CASE 1: SINGLE-STEP TRANSLATION | |
|---|---|
| QUESTION | Are Ramesht and Salajwe both located in the same country? |
| ANSWER | No |
| **TRANSLATION** | |
| ↪QUERY | Are **Ramesht** and **Salajwe** both located in the same country? |
| CLUE | ("**Ramesht**", "## is a village in ##")
("**Salajwe**", "## is a village in ##")
("**Rahim Aga Khan**", "(born ##) is the eldest son of ##") |
| CONTEXT | **1. Salajwe:** Salajwe is a village in Kweneng District of **Botswana**... (Score: 0.64)
**2. Ramesht:** Ramesht ... is a village in ... Kurdistan Province, **Iran**... (Score: 0.63)
**3. Rahim Aga Khan:** Prince Rahim Aga Khan(born 12 October 1971)... (Score: 0.23) |
| PREDICTION | No (✔) |

| CASE 2: MULTI-STEP TRANSLATION | |
|---|---|
| QUESTION | What football club plays in the area between the old tool gates: Brook Bar and Trafford bar? |
| ANSWER | Manchester United F.C. |
| **TRANSLATION** | |
| ↪QUERY | What football club plays in the area between the old tool gates: **Brook Bar** and **Trafford bar**? |
| CLUE | ("**Old Trafford, Greater Manchester**", "## two old toll gates the borders of")
("**Old Trafford, Greater Manchester**", "## is an area of ##, in ##, ##, ## southwest")
("**Brooksbank baronets**", "the ## Baronetcy, of ##, in the parish of ##") |
| CONTEXT | **1. Old Trafford:** ... two old toll gates ... of the area: Brooks's Bar ... and Trafford Bar ... (Score: 0.62)
**2. Brooksbank baronets:** ... of Healaugh Manor, in the parish of Healaugh ... (Score: 0.22) |
| DECOMP. | The evidence provided gives information about Old Trafford and its geographical context, but it does not specify which football club plays in that area.
So the next search query is: What football club is based in Old Trafford, Greater Manchester? |
| ↪QUERY | What football club is based in **Old Trafford, Greater Manchester**? |
| CLUE | ("**Manchester United F.C.**", "## is an ## football club based in ##, ##")
("**Greater Manchester**", "on ## ## south-southwest of ## and ##")
("**Manchester United F.C. in European football**", "## club to enter ## competition") |
| CONTEXT | **1. Manchester United F.C.:** ... is a football club based in Old Trafford, Greater Manchester... (Score: 0.75)
**2. Manchester United F.C. in European football:** ... the first English club to enter... (Score: 0.70)
**3. Middleton, Greater Manchester:** ... on the River Irk 5 mi south-southwest of Rochdale ... (Score: 0.39) |
| PREDICTION | Manchester United F.C. (✔) |

than the passage-level. This is because sentences provide clearer semantic boundaries, which significantly narrows the constraint space for relation decoding. For instance, in 2WikiMQA, passage-level hyperrelations connect an average of 17.2 entities, while sentence-level hyperrelations link only 4.8 entities. Furthermore, during the retrieval stage, we trace clues back to their original passages, due to the translated clues do not contain complete supporting information, and tracing back ensures information integrity. For example, for the question "Who is younger, Ralph Galloway or Antonio Campbell?", one of the clues is: ("Antonio Campbell", "( born ##) is an ## basketball player for ## of ##."), which provides the key semantics "born" related to the birth date of Antonio Campbell, but does not include the specific date.

**The Role of Named Entity Recognition.** The translation process of KG-Translator begins with entities. Accurate entity translation allows the query intent to align well with the local knowledge on ParseKG, thereby defining clear retrieval boundaries. However, NER results often include various entity types (e.g., PERSON, DATE), and not all such types can demarcate precise scopes for subsequent retrieval. For instance, it is clearly unreasonable to demarcate retrieval scopes based on entities such as "12 April 1962" (type: DATE) or "13%" (type: PERCENT), as they do not explicitly point to any specific topic. This renders relational translation ineffective and ultimately leads to the translation of erroneous clues.

With the lightweight NER model, we can easily determine entity types, thereby excluding interferences when constructing the entity-constrained index. We count the interference entities on the large-scale graph, which amount to 145,286 and account for 69.73% of all entities. The large volume of such interference entities and the SOTA performance of KG-Translator indicate that these interference entity types are indeed not the primary medium for aligning queries with the graph.

**The Role of Syntactic Parsing.** In the construction of ParseKG, we utilize syntactic parsing to identify subject-predicate roles, thereby establishing directional hyperrelations from head entities (subjects) to tail entities (non-subjects). Benefiting from this, KG-Translator is able to perform bidirectional translation: forward translation captures semantic relations in the active voice, while reverse translation addresses scenarios where the query intents originate from non-subject entities.

We calculate the contribution of reverse translation across all correctly translated clues, and discover that it accounts for 6.0%, 3.5%, and 11.1% on HotpotQA, 2WikiMQA, and MuSiQue, respectively. This indicates that reverse translation is critical for resolving the intent mismatch between the query and the graph. For instance, given the fact "Bob is the father of Charlie", a query originating from Charlie will trigger reverse translation to generate the clue ("Charlie", "is the father of"). Although this expression misaligns with the relational direction, it successfully anchors the core semantic through the bidirectional mechanism, ensuring the reasoning path remains reachable despite the inversion of relational direction.

Additionally, we further conduct an additional ablation study by replacing directed ParseKG with an undirected version. As shown in Table 16, the undirected version sees a 0.49% drop in average F1 score compared to the directed graph. While the decline is numerically marginal, we argue that using directed graphs remains important for advancing RAG research. On the one hand, the undirected graph causes semantic hyperrelations to degenerate into mere entity co-occurrence, falsely connecting entities like "1981" and "American", which is clearly linguistically nonsensical. On the other hand, since the majority of query intents align with the forward direction, explicit directionality prunes the search space and filters out directionally inconsistent clues during forward translation.

*Table 16.* **F1 score (%) of the undirected ParseKG variant.**

| VARIANT | HQA | 2WIKI | MSQ | AVG. |
|---|---|---|---|---|
| KG-TRANSLATOR (LLAMA) | **80.26** | **78.52** | **43.32** | **68.50** |
| *w* undirected ParseKG | 79.13 | 78.32 | 42.85 | 68.01 |

## I.2. Entity Linking

Entity linking (EL) involves mapping entity mentions within a query to their corresponding entities on a knowledge graph (Hoffart et al., 2011; Le & Titov, 2018; De Cao et al., 2020), serving as an essential process in bridging unstructured language with structured semantic graph. In KG-Translator, precise entity linking (entity translation) is critical, as it directly determines the accuracy of subsequent relation translation and passage retrieval. To further investigate the effectiveness of entity linking in KG-Translator, we divide the entity linking task into two scenarios: (1) **Unambiguous entity linking**: entity mentions can be directly aligned with nodes in the knowledge graph, and (2) **Ambiguous entity linking**: entity mentions are implicit, polysemous, or highly abbreviated, which requires in-depth contextual reasoning for resolution.

We quantitatively analyze these scenarios by calculating $\frac{|\mathcal{E}_c|}{|\mathcal{E}_{\text{total}}|} \times 100\%$ across the successfully linked entities $\mathcal{E}_{\text{total}}$, where $|\mathcal{E}_c|$ denotes the number of category $c$ (unambiguous or ambiguous linking). The results are shown in Figure 10. We obtain the following observation:

**Obs. 25. KG-Translator can effectively link entity mentions in the query to ParseKG, even when such mentions are ambiguous.** Unambiguous entity linking accounts for more than half of the distribution, representing 72.8%, 61.3%, and 66.9% of successful cases on HotpotQA, 2WikiMQA, and MuSiQue, respectively. These distributions reveal that the majority of entity mentions in questions are explicit and can be precisely matched to the entry entities in ParseKG.

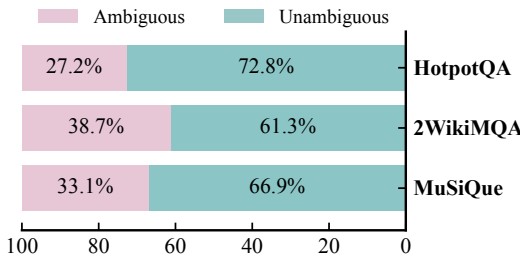

*Figure 10.* **The proportion (%) of ambiguous and unambiguous entity linking across three datasets.**

Meanwhile, KG-Translator achieves an effective linking rate of approximately 30% for ambiguous entity linking. This is because KG-Translator directly translates queries into candidate entities via constrained decoding, which enables full leverage of the semantic comprehension capability of LLMs. For example, for the query "Which agency placed ECT machines in the Class III category?", the term "ECT" is highly abbreviated and inherently ambiguous. Such cases lack explicit semantic overlap with the knowledge graph, posing significant challenges for entity linking. Nevertheless, KG-Translator can capture the contextual semantics within the question, identify the mention accurately and translate it into the correct entity "Electroconvulsive therapy". In summary, KG-Translator bridges the gap between superficial entity mentions and

deep semantics. This ensures precise and robust entity linking across diverse linguistic complexities in real-world scenarios, thereby providing reliable retrieval boundaries for fine-grained clue translation.

### I.3. Semantics

Traditional RAG systems often utilize embedding models to calculate the semantic similarity between queries and candidate passages for retrieval (Gao et al., 2023). In contrast, we advocate translating natural language queries into graph-level clues. From a linguistic perspective, these two paradigms represent distinct levels of semantic carrier: the former relies on the broad contextual background of passages, while the latter focuses on the critical local information within texts. To further demonstrate the semantic advantages of graph-level clues over passage, we conduct quantitative and qualitative evaluations.

We measure the semantic similarity between queries and different semantic carriers. Specifically, we employ all-MiniLM-L6-v2 as the embedding model to map the query $q$, the clue $w$, and the supporting passage $p^*$ into a shared embedding space, where $w$ can uniquely determine $p^*$. Then, we calculate the cosine similarity of $q$ with $w$ and $p^*$, to compare the semantic distances between the two carriers and the query. We define the following two metrics to quantify such distances:

$$\delta_w = \frac{\phi(q) \cdot \phi(w)}{\|\phi(q)\|\|\phi(w)\|}, \quad \delta_{p^*} = \frac{\phi(q) \cdot \phi(p^*)}{\|\phi(q)\|\|\phi(p^*)\|}, \tag{11}$$

where $\phi(\cdot)$ is the embedding function. For each dataset, we compute the average similarity across all successfully retrieved queries, denoted as $\bar{\delta}_w$ and $\bar{\delta}_{p^*}$, respectively. The evaluation results are shown in Table 17, we draw the following observation:

**Obs. 26. Translated clues exhibit a closer distance with the query compared to holistic passages.** $\bar{\delta}_w$ consistently exceeds $\bar{\delta}_{p^*}$, with relative improvements of 6.69%, 11.65%, and 17.00% achieved on the three datasets, respectively. These results stem from the fact that while a holistic passage provides comprehensive contextual information, it inevitably introduces substantial semantic noise irrelevant to the query. In contrast, our clues focus on specific semantic spans, effectively filtering out such semantic noise while preserving the core semantics that align with the query intent. For example, for the query "When

*Table 17.* **Semantic distances between the two carriers and the query.** $\Delta$ represents the relative difference.

| DISTANCE | HQA | 2WIKI | MSQ |
|---|---|---|---|
| PASSAGE ($\bar{\delta}_{p^*}$) | 0.4483 | 0.4457 | 0.3615 |
| CLUE ($\bar{\delta}_w$) | 0.4783 | 0.4976 | 0.4229 |
| $\Delta$ ($\bar{\delta}_{p^*} \to \bar{\delta}_w$) | ↑6.69% | ↑11.65% | ↑17.00% |

did Alexander the Great die?", the essential evidence is within a lengthy passage: "The Hellenistic period covers... between the death of Alexander the Great in 323 BC and the emergence of...". In this scenario, the primary theme of the passage is the "Hellenistic period" rather than the specific event of Alexander's death. The inclusion of extensive irrelevant information introduces substantial contextual noise, yielding a suboptimal semantic similarity score during traditional embedding-based retrieval. Conversely, KG-Translator utilizes translation to obtain the precise clue ("Alexander the Great", "## history between the death of ## in ##"). By generating specific hyperrelation spans, our proposed KG-Translator successfully locates the core semantics from lengthy candidate passages, ensuring more accurate semantic focus.

### I.4. From Clues to Triples

In the knowledge graph field, triples are commonly used to define graph structures for representing independent facts (Luo et al., 2025b). For instance, the sentence "Hugo Maurice Julien Claus was a leading Belgian author" can be represented as ("Hugo Maurice Julien Claus", "is a", "Belgian author"). In contrast, the clues of KG-Translator are expressed as binary tuples, i.e., ("Hugo Maurice Julien Claus", "was a leading ## author"). This difference arises from KG-Translator treating retrieval as a translation task: retrieval requires identifying accurate facts or evidence, while translation only needs to provide a precise and concise retrieval scope. Theoretically, KG-Translator can be extended to a triple-based version. Specifically,

We define the triple clues as $\mathcal{W} = \{(\hat{h}, \hat{r}, \hat{t})\}$, where $\hat{h}, \hat{t}$ denote the translated head and tail entities, respectively, and $\hat{r}$ denotes the translated relation. Therefore, the entire translation consists of three stages: head entity translation, relation translation, and tail entity translation. To ensure the faithfulness of the translation, we first perform KG-constrained Indexing.

**KG-constrained Indexing.** Similarly to Section 3.3.1, we taking forward translation as an example. For head entities and relations, their restriction indices follow Equations 1, 2, and 3, and are stored via a Trie and an FM-index, respectively. For the tail entity restriction index, we first identify the corresponding tail entities based on $\hat{e}$ and $\hat{r}$, a process that can be formally expressed as: $\mathcal{T}^+ = \mathcal{N}_+^{\star}(\hat{e}, \hat{r}) = \mathcal{T}_{\varphi(\hat{e}, \hat{r})}$, where $\mathcal{N}_+^{\star}$ is the entity retrieval function that retrieves all entities pointed to by $\hat{r}$. Considering that $\hat{r}$ is not a hyperrelation but merely a span of a hyperrelation, we need to map the translated

relation back to its corresponding hyperrelation via the mapping function $\varphi$. This process also requires the involvement of the translated entity $\hat{e}$, since $\hat{r}$ is abstract and may occur in multiple hyperrelations. Then, similarly to Equation 1, we tokenize them and store them in Trie to ensure the integrity of the tail entity translation, i.e., $\mathcal{C}_{\mathcal{T}^+} = \text{Trie}(\text{tokenize}(\mathcal{T}^+))$.

**Bidirectional KG Translation.** We modify Equation 4 to enable KG-Translator to forward translate triple clues. Formally,

$$P_\theta(\tau \mid q; \mathcal{G}) = \overbrace{P_h(\hat{\boldsymbol{h}} \mid q; \mathcal{G})}^{\text{Head Translation}} \cdot \overbrace{P_r(\hat{\boldsymbol{r}} \mid q, \hat{\boldsymbol{h}}; \mathcal{G})}^{\text{Relation Translation}} \cdot \overbrace{P_t(\hat{\boldsymbol{t}} \mid q, \hat{\boldsymbol{h}}, \hat{\boldsymbol{r}}; \mathcal{G})}^{\text{Tail Translation}} =$$

$$\prod_{i=1}^{|\hat{\boldsymbol{h}}|} P_\theta(\hat{h}_i \mid q, \hat{h}_{1:i-1}) \cdot \mathcal{C}_{\mathcal{H}}(\hat{h}_i \mid \hat{h}_{1:i-1}) \cdot \prod_{j=1}^{|\hat{\boldsymbol{r}}|} P_\theta(\hat{r}_j \mid q, \hat{\boldsymbol{h}}, \hat{r}_{1:j-1}) \cdot \mathcal{F}_{\mathcal{R}^+}(\hat{r}_j \mid \hat{r}_{1:j-1}) \cdot \tag{12}$$

$$\prod_{k=1}^{|\hat{\boldsymbol{t}}|} P_\theta(\hat{t}_k \mid q, \hat{\boldsymbol{h}}, \hat{\boldsymbol{r}}, \hat{t}_{1:k-1}) \cdot \mathcal{C}_{\mathcal{T}^+}(\hat{t}_k \mid \hat{t}_{1:k-1}),$$

where $P_\theta$ denotes the probability distribution of LLM parameterized by $\theta$, and $\hat{e}_i$, $\hat{r}_j$, and $\hat{t}_k$ respectively denote the $i$-th token of the translated head entity, the $j$-th token of the translated tail entity, and the $k$-th token of the translated relation. $\mathcal{C}_{\mathcal{H}}$ and $\mathcal{F}_{\mathcal{R}^+}$ are constraint functions, which are respectively the same as in Equations 5 and 6. Similarly, $\mathcal{C}_{\mathcal{T}^+}$ is used to constrain KG-Translator to translate a complete tail entity, i.e., $\mathcal{C}_{\mathcal{T}^+}(\hat{t}_k \mid \hat{t}_{1:k-1}) = \mathbb{I}\{\exists \text{prefix}(\hat{t}_{1:k}, \hat{\boldsymbol{t}})\}$.

Furthermore, when the forward translation fails to decode the clues, we activate the reverse translation, which only changes $\mathcal{R}^+$ to $\mathcal{R}^-$ and $\mathcal{T}^+$ to $\mathcal{T}^-$, respectively, where $\mathcal{R}^- = \mathcal{N}_-(\hat{e}) = \{r \mid \hat{e} \in \mathcal{T}_r\}$ and $\mathcal{T}^- = \mathcal{N}_-^{\star}(\hat{e}, \hat{r}) = \mathcal{H}_{\varphi(\hat{e}, \hat{r})}$.

**Structure-aware Fine-tuning.** For the triple setting, we use the constructed training data $\mathcal{O}'$ (see Appendix F.1; we do not perform the final tail entity removal operation) to fine-tune the model. The log-likelihood function is formulated as:

$$\mathcal{L}_{\text{triplet}} = \mathbb{E}_{(q,w) \sim \mathcal{O}'} \log P_\theta(w \mid q)$$

$$= \mathbb{E}\left[\log\left(\prod_{i=1}^{|\hat{\boldsymbol{h}}|} P_\theta(\hat{h}_i \mid q, \hat{h}_{1:i-1}) \cdot \prod_{j=1}^{|\hat{\boldsymbol{r}}|} P_\theta(\hat{r}_j \mid q, \hat{\boldsymbol{h}}, \hat{r}_{1:j-1}) \cdot \prod_{k=1}^{|\hat{\boldsymbol{t}}|} P_\theta(\hat{t}_k \mid q, \hat{\boldsymbol{h}}, \hat{\boldsymbol{r}}, \hat{t}_{1:k-1})\right)\right]. \tag{13}$$

Based on the formulations above, we implement the triple clue translation of KG-Translator. Despite the capability to generate triples, we maintain the binary tuple translation. This setting is grounded in our qualitative analysis of ParseKG's structure and the trade-off between translation efficiency and interpretability. Specifically, we derive two key observations:

**Obs. 27. Triple clues are limited by ParseKG's structural nature, which renders them unable to provide further assistance for retrieval.** While this variant is capable of translating tail entities, we observe that $\hat{t}$ is often unreliable as a factual indicator and redundant for passage retrieval. Since our fine-tuning paradigm aligns the model with ParseKG's structure rather than injecting knowledge, $\hat{t}$ emerges as a product of structural constraints. Specifically, a hyperrelation $\hat{r}$ connects a head entity $\hat{h}$ to a restricted set of entities. This makes the tail entity $\hat{t}$ structurally deterministic given $\hat{h}$ and $\hat{r}$. In some cases, $\hat{t}$ simply represents the only remaining node in the hyperedge. Consequently, $\hat{t}$ relies heavily on the existence of this specific structural path rather than semantic reasoning. Given this structural dependency, we trace the clues back to the original passages to provide reliable context for the general LLMs. In this process, the binary tuple clues are sufficient to uniquely identify the passage candidates, which achieve the same efficacy in passage retrieval with greater conciseness.

**Obs. 28. Triple clues offer a certain degree of interpretability, but they are still insufficient for direct use in QA.** While redundant compared to the binary tuple clues, the triples serve as a more complete semantic abstraction, providing an advantage in terms of human interpretability. To quantify this, we conduct an evaluation using a LLM-based metric termed semantic coverage score (SCS), which is used to determine whether a triple can reflect information in a passage that is beneficial to the query (See Appendix M for prompts). We evaluate on 2,343 clues from 2WikiMQA that can accurately locate supporting passage, and finally achieve an SCS of 69.7%. This result indicates that in some cases, the triples can reflect more complete evidence that captures the facts beneficial to the answer in supporting passages. To illustrate the interpretability of triple clues, we present a case study as shown in Figure 11. For the entity "Peter Emanuel Falck", the clue explicitly extracts the birth date ("15 July 1952") as the tail entity, allowing to verify the answer without reading the full passage. However, in the case of "David Fennario", the translated tail entity corresponds to the place of birth rather than the date. Consequently, while the translated tail entity may be structurally redundant for mechanical retrieval process, the complete triple provides insights for human interpretability.

**Q:** Was David Fennario or Peter Emanuel Falck born first? 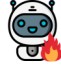

**Triple Clues:**

("David Fennario", "##,( born ##, ##) is a ## playwright best known for ##", "David Wiper")

("Peter Emanuel Falck", "(born ##) is a ## television producer and screenwriter.", "15 July 1952")

*Figure 11.* **A case of triple clues.**

# J. Algorithm of KG-Translator

---

**Algorithm 1** KG Translation

---

**Require:** $q$: User query, $\mathcal{G}$: ParseKG, $\mathcal{M}_\theta$: Trained LLM, $\alpha$: Number of translated entities, $\beta$: Number of translated relations per translated entities, $d$: Translation direction, $k$: Number of supporting passages.
**Ensure:** $\mathcal{W}$: Translated clues, $\hat{\mathcal{P}}_k$: Top-$k$ ranked supporting passages.

1:   **Step 1: Query NER**
2:       $\mathcal{E}_q = \text{NER}(q)$        // Extract query entities $\mathcal{E}_q = \{e_q\}$
3:       $\mathcal{W} = \emptyset$      // Initialize clue set $\mathcal{W}$
4:   **Step 2: Entity Decoding**
5:       $\mathcal{H} = \mathcal{G}.\text{get\_head\_entities}()$        // Get head entities $\mathcal{H}$ as entry
6:       $\mathcal{C}_{\mathcal{H}} = \text{Trie}(\text{tokenize}(\mathcal{H}))$        // Construct a prefix-constrained Trie $\mathcal{C}_{\mathcal{H}}$
7:       $\hat{\mathcal{H}} = \mathcal{M}_\theta(q, \mathcal{E}_q, \mathcal{C}_{\mathcal{H}}, \alpha)$        // Entity decoding, obtaining $\alpha$ translated entities $\hat{\mathcal{H}} = \{\hat{e}\}$
8:   **Step 3: Relation Decoding**
9:       **for** each $\hat{e} \in \hat{\mathcal{H}}$ **do**
10:          $\mathcal{R} = \mathcal{G}.\text{get\_hyperrelations}(\hat{e}, d)$        // Get hyperrelations $\mathcal{R}$ based on direction
11:          $\mathcal{F}_{\mathcal{R}} = \text{FM-index}(\text{tokenize}(\mathcal{R}))$        // Construct a substring-constrained FM-index $\mathcal{F}_{\mathcal{R}}$
12:          $\hat{\mathcal{R}} = \mathcal{M}_\theta(q, \hat{e}, \mathcal{F}_{\mathcal{R}}, \beta)$        // Relation decoding, obtaining $\beta$ translated relations $\hat{\mathcal{R}} = \{\hat{r}\}$
13:          **for** each $\hat{r} \in \hat{\mathcal{R}}$ **do**
14:              $\mathcal{W}.\text{add\_clue}([\hat{e}, \hat{r}])$        // Add clues, where each clue $w = (\hat{e}, \hat{r})$
15:          **end for**
16:      **end for**
17:  **Step 4: Get Top-$k$ Supporting Passages**
18:      $\hat{\mathcal{P}} = \mathcal{G}.\text{get\_passages}(\mathcal{W})$        // Trace back to original passages $\hat{\mathcal{P}}$ based on clues $\mathcal{W}$
19:      $\hat{\mathcal{P}}_k = \arg \text{Top-}k(\text{cosine\_similarity}(\text{emb}(q), \text{emb}(\hat{\mathcal{P}})))$        // Select Top-$k$ ranked passages $\hat{\mathcal{P}}_k$
20:      **Return** $\mathcal{W}, \hat{\mathcal{P}}_k$

---

**Algorithm 2** Bidirectional Translation Mechanism

---

**Require:** $q$: User query, $\mathcal{G}$: ParseKG, $\mathcal{K}$: KG-Translator, $\alpha$: Number of translated entities, $\beta$: Number of translated relations per translated entities, $k$: Number of supporting passages.
**Ensure:** $\mathcal{W}_{\text{all}}$: All translated clues, $\hat{\mathcal{P}}_{\text{all}}$: All ranked supporting passages.

1:   **Step 1: Initialization**
2:       $\mathcal{W}_{\text{all}}, \hat{\mathcal{P}}_{\text{all}} = \emptyset, [\,]$        // Initialize clue set $\mathcal{W}_{\text{all}}$ and passage candidate list $\hat{\mathcal{P}}_{\text{all}}$
3:   **Step 2: Forward Translation**
4:       $\mathcal{W}_+, \hat{\mathcal{P}}_+ = \mathcal{K}.\text{forward\_translate}(q, \mathcal{G}, \alpha, \beta, k)$        // Get clues $\mathcal{W}_+$ and passages $\hat{\mathcal{P}}_+$ via forward translation
5:       $\mathcal{W}_{\text{all}}.\text{add\_clues}(\mathcal{W}_+)$
6:       $\hat{\mathcal{P}}_{\text{all}}.\text{add\_passages}(\hat{\mathcal{P}}_+)$
7:   **Step 3: Reverse Translation**
8:       $\mathcal{E}_q = \text{NER}(q)$        // Identify entities $\mathcal{E}_q$ in the query $q$
9:       $\mathcal{E}_w = \mathcal{W}_{\text{all}}.\text{entites}$        // Identify the translated entities $\mathcal{E}_w$ in the clues $\mathcal{W}$
10:      **if** $\hat{\mathcal{P}}_{\text{all}} = \emptyset$ **or** $\mathcal{E}_q \nsubseteq \mathcal{E}_w$ **then**        // If no retrieval results, or not all NER entities translate into valid clues.
11:          $\mathcal{W}_-, \hat{\mathcal{P}}_- = \mathcal{K}.\text{reverse\_translate}(q, \mathcal{G}, \alpha, \beta, k)$        // Get clues $\mathcal{W}_-$ and passages $\hat{\mathcal{P}}_-$ via reverse translation
12:          $\mathcal{W}_{\text{all}}.\text{add\_clues}(\mathcal{W}_-)$
13:          $\hat{\mathcal{P}}_{\text{all}}.\text{add\_passages}(\hat{\mathcal{P}}_-)$
14:      **end if**
15:  **Step 4: Get Final Clues and Passages**
16:      $\hat{\mathcal{P}}_{\text{all}} = \hat{\mathcal{P}}_{\text{all}}.\text{sort}()$        // Rerank all passages $\hat{\mathcal{P}}_{\text{all}}$
17:      **Return** $\mathcal{W}_{\text{all}}, \hat{\mathcal{P}}_{\text{all}}$

---

## K. Detailed Performance Reports

Given that comprehensive performance reporting across all question types benefits future research, we provide detailed reports covering QA (EM and F1) and retrieval (R@2 and R@5) performance across (1) All baselines and ours KG-Translator (Tables 18 and 19), (2) Ablation study of three variants on KG-Translator (Tables 20 and 21), (3) Various ranking models (Tables 22 and 23), and (4) Different beam sizes (Tables 24 and 25). Notably, key QA data is highlighted in teal, and key retrieval data in maroon; the color definitions are given in each figure's caption.

*Table 18.* **Detailed QA performance of main experiments.** We report the EM (left) and F1 scores (right), where **bold** indicates the best result, underline indicates the second-best result, and the strongest baseline performance is marked in teal.

| METHOD | HOTPOTQA | | 2WIKIMQA | | | | | MUSIQUE | | |
|---|---|---|---|---|---|---|---|---|---|---|
| | BRI.$^{2H}$ | COMP.$^{2H}$ | BRI.$^{2H}$ | COMP.$^{2H}$ | INF.$^{2H}$ | COMP.$^{4H}$ | B.C.$^{4H}$ | BRI.$^{2H}$ | BRI.$^{3H}$ | BRI.$^{4H}$ |
| **VANILLA LLMS** | | | | | | | | | | |
| LLAMA-3.1-8B | 16.2/26.1 | 57.6/62.1 | 5.8/11.8 | 53.6/54.7 | 2.2/25.0 | 48.0/50.1 | 48.8/49.7 | 5.4/14.1 | 3.6/12.1 | 2.9/9.2 |
| GPT-3.5-TRUBO | 31.8/41.1 | 73.4/75.9 | 12.8/18.0 | 52.6/52.8 | 9.2/33.5 | 56.0/56.0 | 46.8/47.4 | 13.4/21.8 | 6.0/15.5 | 5.4/12.2 |
| GPT-4O-MINI | 26.6/35.3 | 70.2/73.7 | 13.0/18.0 | 56.0/56.4 | 3.6/31.3 | 52.8/52.8 | 49.6/50.4 | 10.8/20.5 | 6.0/16.2 | 4.5/14.1 |
| **NAÏVERAG** | | | | | | | | | | |
| NAÏVERAG (TOP-1) | 32.2/41.4 | 72.2/75.1 | 11.4/15.3 | 63.0/63.1 | 4.8/29.1 | 49.6/49.6 | 51.8/52.6 | 12.2/20.6 | 9.8/15.7 | 11.8/17.2 |
| NAÏVERAG (TOP-3) | 45.6/55.3 | 81.8/84.7 | 18.0/22.5 | 78.2/78.3 | 12.4/38.6 | 61.6/61.6 | 64.6/65.0 | 23.0/31.5 | 16.2/25.2 | 19.2/25.7 |
| NAÏVERAG (TOP-5) | 54.2/59.4 | 81.6/84.6 | 20.8/25.3 | 79.6/79.8 | 14.0/40.2 | 60.8/60.8 | 64.0/64.3 | 26.2/35.4 | 19.2/28.1 | 17.9/25.6 |
| **STRUCTURE-AUGMENTED RAG** | | | | | | | | | | |
| KGP | 60.0/72.6 | 75.4/79.3 | 49.0/54.5 | 73.8/74.5 | 34.0/55.7 | 72.8/72.8 | 54.0/54.5 | 31.8/41.7 | 23.2/32.9 | 19.5/27.3 |
| RAPTOR | 45.8/58.4 | 79.4/82.6 | 19.0/24.6 | 73.6/74.2 | 15.6/41.4 | 55.2/55.2 | 57.8/58.3 | 25.0/35.1 | 17.0/26.7 | 16.9/25.7 |
| GRAPHRAG | 30.2/40.1 | 70.8/74.0 | 13.4/16.8 | 62.4/62.6 | 3.8/26.8 | 59.2/59.2 | 49.6/50.3 | 15.2/24.4 | 11.4/21.3 | 4.5/13.6 |
| LIGHTRAG | 55.2/64.8 | 78.0/81.5 | 48.8/56.8 | 78.6/78.7 | 26.6/49.3 | 60.0/60.0 | 64.4/64.7 | 31.8/41.6 | 11.4/23.0 | 11.5/21.1 |
| HIPPORAG | 53.4/66.0 | 84.6/87.5 | 60.6/66.7 | 93.4/93.6 | 35.8/57.5 | 80.8/80.8 | 70.2/70.4 | 33.0/40.9 | 10.0/21.1 | 12.8/22.0 |
| GFM-RAG | 60.4/72.2 | 79.2/82.2 | 66.0/74.0 | 83.4/84.3 | 36.2/56.5 | 74.4/74.4 | 61.0/61.4 | 37.6/46.4 | 19.6/28.3 | 22.7/29.3 |
| HIPPORAG2 | 57.4/69.9 | 82.6/85.7 | 53.6/60.2 | 85.4/85.4 | 30.4/53.6 | 75.2/75.2 | 60.0/60.4 | 39.4/48.4 | 18.0/29.8 | 21.1/27.6 |
| PER-QA | 57.8/69.8 | 81.8/85.1 | 52.4/58.5 | 89.2/89.8 | 42.0/61.0 | 76.0/76.0 | 68.8/68.9 | 43.0/52.2 | 28.4/37.6 | 16.3/25.4 |
| **KG-TRANSLATOR (OURS)** | | | | | | | | | | |
| + LLAMA-3.2-1B | 59.0/72.0 | 85.2/88.5 | 74.6/80.5 | 90.8/90.9 | 53.8/70.7 | 80.0/81.2 | 71.0/71.3 | 46.0/55.5 | 28.8/39.4 | 22.7/30.1 |
| + QWEN-2.5-1.5B | 60.2/72.6 | 86.4/89.5 | 73.4/79.7 | 91.4/91.5 | 52.6/69.1 | 80.8/80.8 | 73.2/73.6 | 47.2/55.7 | 27.0/35.9 | 21.7/29.5 |

*Table 19.* **Detailed retrieval performance of main experiments.** We report the R@2 (left) and R@5 (right), where **bold** indicates the best result, underline indicates the second-best result, and the strongest baseline performance is marked in maroon.

| METHOD | HOTPOTQA | | 2WIKIMQA | | | | | MUSIQUE | | |
|---|---|---|---|---|---|---|---|---|---|---|
| | BRI.$^{2H}$ | COMP.$^{2H}$ | BRI.$^{2H}$ | COMP.$^{2H}$ | INF.$^{2H}$ | COMP.$^{4H}$ | B.C.$^{4H}$ | BRI.$^{2H}$ | BRI.$^{3H}$ | BRI.$^{4H}$ |
| **NAÏVERAG & STRUCTURE-AUGMENTED RAG** | | | | | | | | | | |
| NAÏVERAG (TOP-5) | 53.8/67.5 | 69.0/87.3 | 49.5/56.5 | 64.8/78.3 | 48.8/59.3 | 46.8/59.2 | 34.6/43.1 | 43.8/57.4 | 31.8/44.5 | 19.6/27.9 |
| HIPPORAG | 55.9/75.6 | 85.3/96.2 | 71.2/85.6 | 94.3/97.9 | 58.3/73.9 | 97.6/99.6 | 47.9/86.2 | 45.9/59.0 | 36.6/45.3 | 25.4/32.4 |
| GFM-RAG | 40.1/77.3 | 43.3/80.0 | 59.8/88.6 | 58.6/90.0 | 47.6/80.3 | 61.6/91.6 | 37.5/78.0 | 28.5/54.0 | 19.4/39.2 | 14.9/32.1 |
| HIPPORAG2 | 61.9/80.5 | 68.5/87.2 | 70.0/81.3 | 75.6/87.0 | 60.1/75.3 | 67.2/75.2 | 40.2/60.1 | 53.8/69.1 | 30.3/44.7 | 19.6/28.8 |
| PER-QA | 52.4/67.0 | 70.6/84.5 | 61.9/76.1 | 81.1/91.9 | 55.5/75.8 | 71.2/85.2 | 46.9/72.5 | 50.5/66.1 | 35.3/50.1 | 24.4/32.4 |
| **KG-TRANSLATOR (OURS)** | | | | | | | | | | |
| + LLAMA-3.2-1B | 65.5/76.0 | 83.2/96.6 | 86.3/95.9 | 88.3/95.3 | 80.0/90.5 | 89.6/96.8 | 47.0/87.0 | 52.4/68.7 | 30.8/47.7 | 20.9/31.6 |
| + QWEN-2.5-1.5B | 66.1/76.3 | 84.5/96.4 | 85.7/95.4 | 88.7/96.1 | 80.8/90.5 | 87.6/97.6 | 46.8/86.7 | 53.2/69.0 | 30.2/47.7 | 21.2/32.6 |

*Table 20.* **Detailed QA performance of ablation study.** We report the EM (left) and F1 scores (right) of three variants.

| METHOD | HOTPOTQA | | 2WIKIMQA | | | | | MUSIQUE | | |
|---|---|---|---|---|---|---|---|---|---|---|
| | BRI.$^{2H}$ | COMP.$^{2H}$ | BRI.$^{2H}$ | COMP.$^{2H}$ | INF.$^{2H}$ | COMP.$^{4H}$ | B.C.$^{4H}$ | BRI.$^{2H}$ | BRI.$^{3H}$ | BRI.$^{4H}$ |
| KG-TRANSLATOR | 59.0/72.0 | 85.2/88.5 | 74.6/80.5 | 90.8/90.9 | 53.8/70.7 | 80.0/81.2 | 71.0/71.3 | 46.0/55.5 | 28.8/39.4 | 22.7/30.1 |
| *w/o* structure-aware fine-tuning | 56.6/68.3 | 83.0/86.2 | 55.2/61.0 | 79.4/79.8 | 28.8/50.4 | 75.2/75.8 | 58.0/58.7 | 42.6/49.8 | 26.4/36.7 | 19.5/27.2 |
| *w/o* reverse translation | 53.6/66.1 | 85.2/88.7 | 74.0/80.9 | 91.8/92.0 | 53.0/67.8 | 78.4/79.0 | 73.6/73.9 | 39.4/47.3 | 26.0/34.7 | 16.9/24.7 |
| *w* reverse-then-forward | 50.2/62.2 | 79.4/82.6 | 51.8/57.6 | 90.8/90.8 | 30.0/52.5 | 79.2/79.2 | 69.6/69.8 | 40.2/48.3 | 22.0/32.0 | 15.3/24.0 |

*Table 21.* **Detailed retrieval performance of ablation study.** We report the R@2 (left) and R@5 (right) of three variants.

| METHOD | HotpotQA | | 2WikiMQA | | | | | MuSiQue | | |
|---|---|---|---|---|---|---|---|---|---|---|
| | Bri.[2H] | Comp.[2H] | Bri.[2H] | Comp.[2H] | Inf.[2H] | Comp.[4H] | B.C.[4H] | Bri.[2H] | Bri.[3H] | Bri.[4H] |
| KG-Translator | 65.5/76.0 | 83.2/96.6 | 86.3/95.9 | 88.3/95.3 | 80.0/90.5 | 89.6/96.8 | 47.0/87.0 | 52.4/68.7 | 30.8/47.7 | 20.9/31.6 |
| *w/o* structure-aware fine-tuning | 61.2/69.9 | 79.0/87.1 | 69.7/73.9 | 75.7/78.2 | 59.9/64.0 | 89.2/75.2 | 32.1/49.8 | 51.3/63.3 | 30.0/44.1 | 20.0/32.8 |
| *w/o* reverse translation | 60.4/65.9 | 85.7/97.0 | 88.1/95.9 | 90.9/96.1 | 80.8/88.9 | 91.2/98.8 | 48.4/91.3 | 48.4/58.6 | 31.1/46.7 | 20.1/31.2 |
| *w* reverse-then-forward | 52.1/60.1 | 52.8/61.7 | 67.3/73.6 | 84.4/93.2 | 52.8/60.8 | 84.0/96.8 | 45.6/79.0 | 46.9/62.3 | 27.7/40.3 | 18.6/28.9 |

*Table 22.* **Detailed QA performance of different ranking models.** We report the EM (left) and F1 scores (right), where **bold** indicates the best result, underline indicates the second-best result. The performance to the default model for each dataset is marked in teal.

| MODEL | HotpotQA | | 2WikiMQA | | | | | MuSiQue | | |
|---|---|---|---|---|---|---|---|---|---|---|
| | Bri.[2H] | Comp.[2H] | Bri.[2H] | Comp.[2H] | Inf.[2H] | Comp.[4H] | B.C.[4H] | Bri.[2H] | Bri.[3H] | Bri.[4H] |
| all-MiniLM-L6-v2 | 59.0/72.0 | 85.2/88.5 | 74.6/80.5 | 90.8/90.9 | 53.8/70.7 | 80.0/81.2 | 71.0/71.3 | 46.0/55.5 | 28.8/39.4 | 22.7/30.1 |
| all-mpnet-v2 | 59.6/71.9 | 85.2/88.7 | 74.8/80.2 | 89.6/90.2 | 50.8/68.6 | 81.6/81.6 | 74.2/74.3 | 45.0/52.8 | 30.4/39.5 | 19.8/27.5 |
| bge-large-en-v1.5 | 61.6/74.2 | 85.8/88.7 | 75.0/80.7 | 90.6/90.8 | 53.6/70.0 | 83.2/83.2 | 74.6/75.0 | 46.2/53.7 | 30.6/40.4 | 23.3/30.5 |
| gte-Qwen2-1.5b-instruct | 59.8/71.1 | 84.6/88.1 | 73.8/80.2 | 89.8/90.0 | 52.4/69.0 | 79.2/79.2 | 73.0/73.3 | 44.6/52.2 | 29.8/39.2 | 17.9/25.8 |

*Table 23.* **Detailed retrieval performance of different ranking models.** We report the R@2 (left) and R@5 (right), where **bold** indicates the best result, underline indicates the second-best result. The performance to the default model for each dataset is marked in maroon.

| MODEL | HotpotQA | | 2WikiMQA | | | | | MuSiQue | | |
|---|---|---|---|---|---|---|---|---|---|---|
| | Bri.[2H] | Comp.[2H] | Bri.[2H] | Comp.[2H] | Inf.[2H] | Comp.[4H] | B.C.[4H] | Bri.[2H] | Bri.[3H] | Bri.[4H] |
| all-MiniLM-L6-v2 | 65.5/76.0 | 83.2/96.6 | 86.3/95.9 | 88.3/95.3 | 80.0/90.5 | 89.6/96.8 | 47.0/87.0 | 52.4/68.7 | 30.8/47.7 | 20.9/31.6 |
| all-mpnet-v2 | 66.8/76.6 | 87.6/96.9 | 86.3/95.8 | 89.2/95.4 | 77.6/90.5 | 92.0/98.4 | 47.1/88.3 | 54.5/68.4 | 31.1/49.3 | 54.5/31.7 |
| bge-large-en-v1.5 | 70.3/77.5 | 93.0/97.6 | 93.9/96.2 | 93.6/96.4 | 86.2/90.3 | 96.8/98.8 | 49.5/92.8 | 56.4/71.5 | 32.7/52.5 | 22.0/36.0 |
| gte-Qwen2-1.5b-instruct | 65.5/76.4 | 84.0/96.3 | 86.1/95.9 | 84.4/96.0 | 72.5/89.8 | 92.4/98.0 | 46.4/87.5 | 50.5/66.2 | 28.2/44.8 | 17.2/29.9 |

*Table 24.* **Detailed QA performance of different beam sizes.** We report the EM (left) and F1 scores (right), where **bold** indicates the best result, underline indicates the second-best result. The performance to the hyperparameters for each dataset is marked in teal.

| BEAM SIZE | HotpotQA | | 2WikiMQA | | | | | MuSiQue | | |
|---|---|---|---|---|---|---|---|---|---|---|
| | Bri.[2H] | Comp.[2H] | Bri.[2H] | Comp.[2H] | Inf.[2H] | Comp.[4H] | B.C.[4H] | Bri.[2H] | Bri.[3H] | Bri.[4H] |
| $(\alpha=1, \beta=1)$ | 55.8/66.3 | 85.4/88.7 | 71.2/77.6 | 92.4/92.6 | 40.2/60.7 | 79.2/79.8 | 75.0/75.3 | 41.6/49.2 | 17.8/26.6 | 16.0/21.7 |
| $(\alpha=1, \beta=3)$ | 55.8/68.1 | 85.6/89.0 | 71.4/77.6 | 91.8/92.0 | 42.6/63.5 | 80.8/80.8 | 72.4/72.9 | 44.8/52.9 | 22.0/31.6 | 16.9/23.5 |
| $(\alpha=1, \beta=5)$ | 56.2/68.6 | 84.6/88.2 | 72.0/77.8 | 92.8/93.0 | 44.8/64.7 | 78.4/78.4 | 72.0/72.3 | 46.8/54.9 | 23.6/33.4 | 16.0/23.6 |
| $(\alpha=3, \beta=1)$ | 58.0/70.6 | 85.0/88.8 | 74.6/80.5 | 90.8/90.9 | 53.8/70.7 | 80.0/81.2 | 71.0/71.3 | 43.8/51.5 | 25.2/34.5 | 18.5/24.2 |
| $(\alpha=3, \beta=3)$ | 59.0/72.0 | 85.2/88.5 | 73.8/80.5 | 92.0/92.1 | 47.6/66.0 | 77.6/77.6 | 70.8/71.3 | 46.8/55.4 | 28.4/37.6 | 16.3/25.1 |
| $(\alpha=3, \beta=5)$ | 57.6/70.1 | 85.4/88.7 | 73.8/80.3 | 91.2/91.3 | 46.6/65.9 | 80.0/80.6 | 70.8/71.0 | 46.0/55.5 | 28.8/39.4 | 22.7/30.1 |
| $(\alpha=5, \beta=1)$ | 58.8/70.8 | 85.4/88.5 | 74.2/80.7 | 91.0/91.2 | 55.0/70.9 | 77.6/78.2 | 68.8/69.0 | 46.2/53.6 | 24.2/33.4 | 15.3/22.5 |
| $(\alpha=5, \beta=3)$ | 59.0/71.4 | 85.4/88.6 | 74.8/81.1 | 89.2/89.4 | 51.2/69.0 | 81.6/82.2 | 68.6/69.0 | 45.2/53.7 | 29.0/37.7 | 19.2/26.6 |
| $(\alpha=5, \beta=5)$ | 59.6/71.7 | 85.8/88.9 | 73.2/79.7 | 91.8/91.9 | 47.2/66.2 | 80.8/81.3 | 72.0/72.1 | 45.2/54.2 | 28.6/39.0 | 19.5/27.5 |

*Table 25.* **Detailed retrieval performance of different beam sizes.** We report the R@2 (left) and R@5 (right), where **bold** indicates the best result, underline indicates the second-best result. The performance to the hyperparameters for each dataset is marked in maroon.

| BEAM SIZE | HotpotQA | | 2WikiMQA | | | | | MuSiQue | | |
|---|---|---|---|---|---|---|---|---|---|---|
| | Bri.[2H] | Comp.[2H] | Bri.[2H] | Comp.[2H] | Inf.[2H] | Comp.[4H] | B.C.[4H] | Bri.[2H] | Bri.[3H] | Bri.[4H] |
| $(\alpha=1, \beta=1)$ | 58.4/60.5 | 85.2/85.7 | 86.1/88.0 | 92.1/93.3 | 66.7/68.8 | 95.2/97.6 | 48.1/87.6 | 51.0/60.6 | 29.3/39.4 | 21.2/29.6 |
| $(\alpha=1, \beta=3)$ | 61.2/66.7 | 84.2/90.6 | 86.9/91.9 | 91.4/95.5 | 71.2/78.8 | 91.2/97.6 | 47.2/87.7 | 52.1/65.6 | 29.0/41.3 | 20.4/30.4 |
| $(\alpha=1, \beta=5)$ | 61.7/67.9 | 83.5/90.4 | 87.1/92.8 | 92.1/96.3 | 73.1/82.0 | 91.6/97.2 | 47.9/88.3 | 52.0/66.7 | 29.7/43.2 | 20.1/31.4 |
| $(\alpha=3, \beta=1)$ | 65.5/72.8 | 85.3/95.0 | 86.3/95.9 | 88.3/95.3 | 80.0/90.5 | 89.6/96.8 | 47.0/87.0 | 51.9/65.3 | 31.3/47.3 | 19.5/32.2 |
| $(\alpha=3, \beta=3)$ | 65.5/76.0 | 83.2/96.6 | 84.4/95.3 | 88.5/97.3 | 74.7/88.4 | 88.4/98.0 | 46.9/86.1 | 51.1/68.4 | 30.7/47.7 | 20.9/31.4 |
| $(\alpha=3, \beta=5)$ | 65.1/75.8 | 82.9/96.4 | 83.2/95.6 | 87.2/95.9 | 73.8/88.1 | 88.0/97.6 | 46.8/85.5 | 52.4/68.7 | 30.8/47.7 | 20.9/31.6 |
| $(\alpha=5, \beta=1)$ | 64.4/73.1 | 83.5/95.5 | 86.3/96.2 | 87.1/95.7 | 79.2/91.1 | 86.8/96.8 | 47.0/85.5 | 53.0/66.5 | 30.9/47.3 | 19.6/32.2 |
| $(\alpha=5, \beta=3)$ | 65.0/75.2 | 82.1/96.5 | 84.3/95.9 | 84.6/95.5 | 75.2/89.8 | 86.4/98.4 | 46.5/84.0 | 52.1/67.6 | 29.9/47.9 | 20.4/31.3 |
| $(\alpha=5, \beta=5)$ | 65.8/76.3 | 81.2/96.7 | 83.0/95.1 | 86.4/97.6 | 73.6/88.5 | 86.0/98.0 | 46.5/84.8 | 51.7/68.4 | 30.3/48.1 | 19.9/32.1 |

# L. Detailed Baseline Configuration

In our experiments, we provide detailed implementation specifications for structure-augmented RAG baselines to ensure a fair comparison. As shown in Table 26: (1) **Embedding** denotes the embedding model (retriever); (2) **Chunking** indicates the chunking size (original denotes passages in original datasets); (3) **Ctx.** specifies the number of retrieved context passages (Top-$k$); (4) **Ent.** denotes the number of retrieved entities (Top-$k$); (5) **Steps** represents the number of retrieval or iterative steps; and (6) **Other Important Hyperparameters** includes key hyperparameters for the reproduction of each method.

*Table 26.* **Detailed hyperparameter configurations for structure-augmented RAG baselines.**

| METHOD | EMBEDDING | CHUNKING | CTX. | ENT. | STEPS | OTHER IMPORTANT HYPERPARAMETERS |
|---|---|---|---|---|---|---|
| KGP | all-MiniLM-L6-v2 | original | 5 | – | 2 | "initial_retrieval_top_k": 5, "max_neighbors": 5, "linking_threshold": 0.8, "summary_max_tokens": 500 |
| RAPTOR | all-MiniLM-L6-v2 | original | 5 | – | 1 | "num_layers": 5, "threshold": 0.1, "threshold_cluster_num": 5000, "max_length_in_cluster": 3500 |
| GRAPHRAG | all-MiniLM-L6-v2 | 300 | – | 10 | 1 | "retrieval_mode": "local", "max_cluster_size": 10, "community_reports_max_length": 2000, "community_reports_max_input_length": 8000 |
| LIGHTRAG | all-MiniLM-L6-v2 | 1200 | – | 5 | 1 | "retrieval_mode": "local", "max_token_for_text_unit": 4000, "max_entity_tokens": 6000, "max_relation_tokens": 8000 |
| HIPPORAG | all-MiniLM-L6-v2 | original | 5 | – | 1 | "graph_type": "facts_and_sim", "synonymy_threshold": 0.8, "graph_algorithm": "ppr", "damping": 0.5 |
| GFM-RAG | all-mpnet-v2 (trained) | original | 5 | – | 2/4 | "threshold": 0.8, "max_sim_neighbors": 100, "doc_ranker": "idf_topk_ranker", "doc_ranker_top_k": 70 |
| HIPPORAG2 | all-MiniLM-L6-v2 | original | 5 | – | 1 | "graph_type": "facts_and_sim", "synonymy_threshold": 0.8, "graph_algorithm": "ppr", "damping": 0.5 |
| PER-QA | all-MiniLM-L6-v2 | original | 5 | – | 1-8 | "type_classification": true, "planner_mode": "universal", "executor_mode": "rag", "reasoner_mode": "direct" |

# M. Prompts

We provide all the prompts used in the experiments, as shown in Figure 12. KG-Translator accomplishes the RAG task in two stages: retrieval and QA. For retrieval, we design the prompt for the model's training (structure-aware fine-tuning) and translation. For QA, we design a JSON-formatted prompt that facilitates result parsing by the program. We unify the QA prompt for all baselines to this one to ensure fair comparison of results. Furthermore, we also provide question decomposition prompts for multi-hop settings and triple clue translation prompts. For question decomposition, we design a prompt with chain-of-thought (Wei et al., 2022), which generates a new search query when the current context is insufficient to answer the question. For triple translation, we revise the original translation prompt to adapt it for triple clue translation.

In addition, we provide the prompt for evaluating semantic coverage score (SCS) that is mentioned in Section I.4.

---

**❶ Clue Translation Prompt (Training and Inference):**
Given a query, your task is to provide evidence in a pair format to help answer the query. A pair consists of a entity and a relation,, which are connected by "@@" and follows the format of "ENTITY@@RELATION".

**❷ QA Prompt:**
Given a question and its evidence, your task is to answer the question based on the evidence. You must give ONLY the direct answer in the most concise way possible.
Generate a JSON with a single key "Response" and a value that is a short phrase or a few words. In JSON, put every value as a string always, not float. Format your response as follows: {{"Response": "ANSWER_HERE"}}

**❸ Question Decomposition Prompt:**
Given a question and its evidence, your task is to generate next search query different from the used queries to obtain additional evidence. First conduct step-by-step thinking, then start with "So the next search query is" and generate a concise search query that can supplement this missing content.
1. If the evidence is insufficient to answer the question: Identify the missing information based on the key information extracted, and generate a concise search query that can supplement this missing content.
2. If the evidence is sufficient to fully answer the question: Clearly state the final answer to the original question in the thinking process, and generate "None" as the next search query.

**❹ Triple Clue Translation Prompt (Training and Inference):**
Given a query, your task is to provide evidence in a triple format to help answer the query. A triple consists of a head entity, a relation, and a tail entity, which are connected by "@@" and follows the format of "HEAD_ENTITY@@RELATION@@TAIL_ENTITY".

**❺ Semantic Coverage Score Prompt:**
You are an expert evaluator in information retrieval. Your task is to judge whether a given clue serves as a reasonable semantic anchor that justifies retrieving the passage for the query.

** Criteria for Score 1: **
- Entity anchoring: The clue identifies the main entity discussed in the query.
- Semantic relevance: The clue contains keywords that match the query's topic.
- Partial coverage: Even if the clue misses the answer, it scores 1 for pointing to the correct information segment.

** Criteria for Score 0: **
- Total irrelevance: The clue contains NO keywords related to the query.
- Pure noise: The clue consists only of stopwords, punctuation, or generic verbs.
- Hallucination: The clue contains specific entities that do not appear in the passage.

Output strictly in JSON format: {"score": 1} or {"score": 0}.

---

*Figure 12.* **All prompts used in this paper.**

