# OpenReview forum: "From Retrieval to Translation: Translating Query into Graph-level Clues for Retrieval-Augmented Generation"
_ICML.cc/2026/Conference — ICML 2026 regular_

### Official Review · Reviewer_5AAG · 2026-03-08

**Soundness:** 2
**Presentation:** 3
**Significance:** 2
**Originality:** 2
**Overall Recommendation:** 4
**Confidence:** 4

**Summary:**

This paper introduces KG-Translator, a RAG paradigm that translates user queries into graph-level clues rather than performing traditional embedding-based retrieval. It constructs a parsing-based knowledge graph (ParseKG) using lightweight NLP models at zero token cost, then applies constrained decoding (Trie + FM-index) to faithfully translate queries into entity-relation clues that are traced back to original passages. Motivated by retrieval suspension and semantic drift issues in existing structure-augmented RAG, KG-Translator with 1B-parameter backbones achieves ~6.7% average improvements over baselines across four multi-hop QA benchmarks with significantly lower construction costs.

**Compliance With Llm Reviewing Policy:**

Affirmed.

**Key Questions For Authors:**

**Q1.** Since the performance ceiling appears tied to ParseKG quality rather than model capacity, what is the viable path for improvement when spaCy's general-purpose NER becomes insufficient, for instance in specialized domains (e.g., biomedical, legal) where domain-specific entities may not be recognized? Does this require redesigning the entire ParseKG construction pipeline per domain?

**Q2.** Could you provide results where these baselines are equipped with larger embedding models (e.g., GTE-Qwen2-7B), as was done for NaïveRAG in Table 11, to isolate whether the gains stem from the translation paradigm itself or from suboptimal baseline configurations?

**Q3.** Could you provide an end-to-end inference time comparison against baselines (e.g., HippoRAG2, GFM-RAG), particularly on the large-scale ScaleQA corpus?

**Limitations:**

“Yes”

**Strengths And Weaknesses:**

**Strengths**

The paper clearly identifies limitations in existing structure-augmented RAG approaches and attempts to address them through a constrained decoding strategy. The experimental results demonstrate that the proposed method achieves strong performance even with a relatively small model. In addition, the authors provide extensive comparative and ablation studies, which enhance the clarity, reproducibility, and overall understanding of the proposed approach.


**Weaknesses**

**W1. Performance Ceiling Due to Model Scaling**

According to Table 10, scaling up the backbone model from 1B to 3B yields almost no performance improvement, and in some cases even leads to degradation. This suggests that because constrained decoding restricts the LLM's output to ParseKG, the performance ceiling is determined by the quality of ParseKG rather than by model capacity, structurally blocking the path to performance improvement through model scaling. This could become a serious limitation when extending the method to more complex scenarios such as specialized domains.

**W2. Potential Underestimation of Baselines Due to the Use of a Small Embedding Model**

In the paper, both the baseline methods and the proposed method use the small embedding model all-MiniLM-L6-v2 (22.7M) for retrieval. According to Table 4, increasing the embedding model size has little effect on the performance of KG-Translator; however, structure-augmented RAG baselines such as GRAPHRAG, LIGHTRAG, and HIPPORAG are methods that heavily rely on semantic embeddings. Using a small embedding model for these methods may prevent them from fully leveraging semantic information, thereby underestimating their actual performance. Table 11 only presents a comparison using larger embedding models in the NaïveRAG setting, while an evaluation of structure-augmented RAG baselines with appropriately sized embedding models is absent. To more convincingly demonstrate the superiority of the proposed method, a fair comparison under conditions where embedding-dependent approaches can perform at their full potential is necessary.

**W3. Insufficient Verification of the Practical Effectiveness and Inference Cost of Constrained Decoding**

The authors apply constrained decoding to address two problems of existing structure-augmented RAG (retrieval suspension and semantic drift); however, since Trie- and FM-index-based constrained decoding must be performed dynamically during inference, additional latency may be introduced. Although a component-wise time analysis is provided in Table 9, it is limited to only two question types from 2WikiMQA. A direct comparison of end-to-end inference time on real large-scale corpora or a head-to-head inference speed comparison against existing baselines is absent.

---

> ### Author Rebuttal · Authors · 2026-03-30
>
> Dear 5AAG,
>
> We sincerely appreciate the time and efforts you have devoted to identifying potential issues in our paper. We are also grateful for the opportunity to further clarify the design motivations and experiment details of our framework, which helps us refine and improve the work accordingly.
>
> ---
>
> ### W1 & Q1: Regarding the optimization potential of KG-Translator and ParseKG
>
> ### (1) Scaling law of KG-Translator (W1)
>
> Thanks for your professional suggestions. We provide detailed experimental data to clarify the scaling law of KG-Translator in detail.
>
> *Table 1: EM of Backbone Model Scaling on Different Question Types*
>
> |Backbone Model|HotpotQA Bri.$^\text{2H}$|MuSiQue Bri.$^\text{2H}$|MuSiQue Bri.$^\text{3H}$|MuSiQue Bri.$^\text{4H}$|
> |-|-|-|-|-|
> |Llama-1B|59.00|46.00|28.80|22.68|
> |Llama-3B|60.20|47.60|29.00|27.80|
> |$\Delta$1B to 3B (Llama)|***+1.20***|***+1.60***|***+0.20***|***+5.12***|
> |Qwen-1.5B|60.20|47.20|27.00|21.73|
> |Qwen-3B|60.80|45.40|29.00|24.28|
> |$\Delta$1.5B to 3B (Qwen)|***+0.60***|*-1.80*|***+2.00***|***+2.55***|
>
> As shown in Table 1, we observe significant performance gains from model scaling on the 4-hop Bridge questions of MuSiQue: 5.12% for Llama and 2.55% for Qwen. We analyze this phenomenon in detail:
>
> + Question type: Compared with 2-hop questions, 4-hop questions require KG-Translator to have stronger translation capabilities to filter out irrelevant clues.
> + Corpus: Compared with HotpotQA, MuSiQue contains more passages on the same topic. For instance, there is only 1 passage about "Einstein" in HotpotQA but 5 in MuSiQue.
>
> Therefore, **KG-Translator still follows the scaling law under model scaling on more complex question types and corpora**, indicating its performance is not fully constrained by ParseKG. Meanwhile, the quality of ParseKG remains critical for accurate translation.
>
> ### (2) Practical improvement paths of ParseKG (Q1)
>
> Thanks for your expertise. In the design of ParseKG, **components such as NER and syntactic parsing are decoupled**. We can thus replace the spaCy-based NER with a more powerful LLM-based NER or domain-specialized NER to **improve NER quality without affecting subsequent workflows**.
>
> We conduct additional experiments with LLM‑based NER as follows:
>
> *Table 2: Retrieval Performance of KG-Translator on 2WikiMQA Inf.$^\text{2H}$*
>
> |Method|Recall@2|Recall@5|
> |-|-|-|
> |KG-Translator (spaCy-based NER)|80.4|92.5|
> |KG-Translator (LLM-based NER)|81.2 (+0.8)|93.1 (+0.6)|
>
> As shown in Table 2, **optimizing ParseKG with LLM-based NER improves performance**, demonstrating its optimization potential.
>
> ---
>
> ### W2 & Q2: Regarding the Structure-Augmented RAG with large embedding models
>
> Thanks for your valuable feedback. We conduct supplementary experiments by equipping the strong baseline HippoRAG with various large embedding models (LEMs).
>
> *Table 3: F1 scores of Structure-Augmented RAG with LEMs*
>
> |Method|Embedding Model|HotpotQA|2WikiMQA|MuSiQue|Avg.|
> |-|-|-|-|-|-|
> |HippoRAG|all-MiniLM-L6-v2 (22.7M)|76.80|72.54|28.86|60.58|
> |HippoRAG|GTE-Qwen2-7B (7B)|75.88|69.28|26.89|58.23|
> |HippoRAG|NV-Embed-v2 (7B)|77.48|72.40|29.41|60.83|
> |KG-Translator|all-MiniLM-L6-v2 (22.7M)|**80.26**|**78.52**|**43.32**|**68.50**|
>
> As shown in Table 3, even when employing LEMs, **Structure-Augmented RAG still exhibits significantly lower performance than KG-Translator**.
>
> ---
>
> ### W3 & Q3: Regarding the efficiency
>
> Thanks for your considerate suggestions. Following your suggestion, we add the component-wise and end-to-end time analyses on the **large-scale corpus ScaleQA** and **other baselines**. In addition, we further optimize our code using caching techniques. We will include the updated results in the final version.
>
> *Table 4: Component-wise Retrieval Time of KG-Translator*
>
> |Time (s)|2WikiMQA (5k)|ScaleQA (35k)|
> |-|-|-|
> |NER|0.067 (5.2%)|0.067 (2.8%)|
> |Entity-constrained Indexing (Trie)|0.455 (35.3%)|1.324 (56.0%)|
> |Entity Translation|0.173 (13.4%)|0.173 (7.3%)|
> |Relation-constrained Indexing (FM-Index)|0.021 (1.6%)|0.071 (3.0%)|
> |Relation Translation|0.556 (43.2%)|0.590 (24.9%)|
> |Ranking|0.016 (1.2%)|0.141 (5.9%)|
> |Total Time (s)|1.288|2.367|
>
> As shown in Table 4, as the corpus scales up, **the latency of most components shows no significant changes**.
>
> *Table 5: Full-Stage Efficiency Comparison with Baselines on ScaleQA. "QD": question decomposition*
>
> |Method|Total Indexing (s)|Avg. Retrieval (s)|Avg. QD (s)|QD Num.|Avg. QA (s)|F1 (%)|
> |-|-|-|-|-|-|-|
> |HippoRAG2|12344.30|4.68|-|-|1.69|54.85|
> |GFM-RAG|17820.10|1.30|1.09|2-4|1.73|64.04|
> |KG-Translator|3234.72|2.37|2.13|2-4|1.66|68.33|
>
> As shown in Table 5, KG-Translator **significantly reduces indexing time**. While retrieval latency rises slightly, this is acceptable given the improved QA performance.
>
> ---
>
> We once again express our sincere appreciation for your time, efforts, and thoughtful feedback. If you have any additional comments or suggestions, we would greatly value your comments.
>
> Best regards!

---

> > ### Author Rebuttal · Reviewer_5AAG · 2026-04-01
> >
> > Thank you to the authors for the thorough rebuttal. The responses to W1–W3 are satisfactory, and I am raising my score from 3 to 4 accordingly.
> >
> > Separately, regarding W2, Table 3 shows that equipping HippoRAG with larger embedding models yields no meaningful improvement or even degrades performance. This is counterintuitive, as I would expect larger embeddings to benefit methods that heavily rely on semantic similarity. Is this a known phenomenon in graph-based RAG, and what explains it?

---

> > > ### Author Response · Authors · 2026-04-02
> > >
> > > Dear 5AAG,
> > >
> > > Thanks for your professional feedback. We are pleased to note that your concerns have been addressed, and we appreciate your insightful follow-up questions.
> > >
> > > ---
> > >
> > > ### Regarding further analysis on W2
> > >
> > > Thanks for your valuable suggestions. Large embedding models (LEMs) may bring no significant improvement or even performance degradation. The main reasons are as follows:
> > >
> > > In RAG systems, large embedding models act as retrievers and are mainly fine-tuned on passage-level retrieval tasks, which equips them with strong semantic understanding capabilities [1].  In structure-augmented RAG systems, methods like HippoRAG **employ embedding models for entity linking**, i.e., mapping query entities to graph entities [2, 4]. However, when applied to entity linking tasks, large embedding models cannot fully leverage their strong semantic understanding ability, as **entities lack sufficient background knowledge**. For instance, the entity "apple" is hard to disambiguate without context such as "fruit" or "smartphone brand". In other words, **entity linking is essentially a word-matching task simpler than passage-level retrieval**, thus leading to insignificant performance differences among different-scale embedding models.
> > >
> > > This phenomenon is also observed in published papers, including HippoRAG and GFM-RAG.
> > >
> > > *Table 6: Recall@5 of HippoRAG with Different Embedding Models (Excerpted from Table 2 of HippoRAG [2] and Table 3 of HippoRAG2 [3])*
> > >
> > > | Method   | Embedding Model  | HotpotQA | 2WikiMQA | MuSiQue |
> > > | -------- | ---------------- | -------- | -------- | ------- |
> > > | HippoRAG | ColBERTv2 (0.1B) | 77.7     | 89.1     | 51.9    |
> > > | HippoRAG | NV-Embed-v2 (7B) | 77.3     | 90.4     | 53.2    |
> > >
> > > *Table 7: Recall@5 of GFM-RAG with Different Embedding Models (Excerpted from Table 8 of GFM-RAG [4])*
> > >
> > > | Method  | Embedding Model          | HotpotQA | 2WikiMQA | MuSiQue |
> > > | ------- | ------------------------ | -------- | -------- | ------- |
> > > | GFM-RAG | all-mpnet-base-v2 (0.1B) | 82.1     | 85.6     | 55.1    |
> > > | GFM-RAG | NV-Embed-v2 (7B)         | 81.4     | 85.5     | 54.9    |
> > >
> > > As shown in Tables 6 and 7, equipping HippoRAG and GFM-RAG with large embedding models brings no meaningful improvement or even degrades performance across all three datasets, which is consistent with our results in W2.
> > >
> > > In summary, large embedding models do not always bring performance gains to structure-augmented RAG systems. Additionally, since retrieval tasks demand high efficiency, small-scale embedding models (e.g., Qwen3-Embedding-0.6B) are widely used in industry. Our proposed translation paradigm follows the same practice and achieves a favorable balance between efficiency and performance with small embedding models.
> > >
> > > + **Reference**
> > >
> > > [1] NV-Embed: Improved Techniques for Training LLMs as Generalist Embedding Models. (Lee et al., ICLR'25)
> > >
> > > [2] HippoRAG: Neurobiologically Inspired Long-term Memory for Large Language Models. (Gutiérrez et al., NeurIPS'24)
> > >
> > > [3] From RAG to Memory: Non-parametric Continual Learning for Large Language Models. (Gutiérrez et al., ICML'25)
> > >
> > > [4] GFM-RAG: Graph Foundation Model for Retrieval Augmented Generation. (Luo et al., NeurIPS'25)
> > >
> > > ---
> > >
> > > Thank you again sincerely for taking the time to review our paper. We sincerely appreciate your constructive feedback and your positive evaluation of our work.
> > >
> > > Best regards!

---

### Official Review · Reviewer_Rc27 · 2026-03-08

**Soundness:** 2
**Presentation:** 3
**Significance:** 2
**Originality:** 2
**Overall Recommendation:** 4
**Confidence:** 4

**Summary:**

This paper presents KG-Translator, a retrieval framework that reformulates structure-augmented RAG as a query-to-clue translation problem. Instead of relying purely on graph matching, it constructs a parsing-based graph, translates questions into structured clues with constrained decoding, and traces them back to relevant evidence. The approach is well motivated, and the experiments show strong improvements over competitive baselines on multi-hop QA benchmarks.

**Compliance With Llm Reviewing Policy:**

Affirmed.

**Final Justification:**

The authors have addressed my concerns.

**Key Questions For Authors:**

Please address the questions in the weaknesses.

**Limitations:**

yes

**Strengths And Weaknesses:**

Strengths:

1. The paper is well motivated and identifies clear limitations in prior graph-based RAG methods.
2. The overall framework is intuitive and technically well organized.
3. The empirical results are strong, with both effectiveness and efficiency gains.

Weaknesses

1. The method appears to rely heavily on the quality of the parsing pipeline, especially NER and syntactic analysis. If these upstream components make mistakes, the constructed graph may be incomplete or noisy, and the paper does not fully analyze how robust the framework is under such conditions.

2. It is somewhat difficult to isolate where the performance gains truly come from.

3. The conceptual novelty feels somewhat limited. Many of the individual components are familiar, and the main contribution seems to come from combining them effectively rather than introducing a fundamentally new retrieval paradigm.


4. The efficiency results are promising, but they are not entirely conclusive as presented. A more complete end-to-end latency comparison, including all major stages of the pipeline, would make the practical advantage of the method more convincing.

---

> ### Author Rebuttal · Authors · 2026-03-30
>
> Dear Rc27,
>
> We sincerely appreciate your devoting significant time and efforts to providing valuable insights and constructive comments. To address your concerns, we conduct rigorous supplementary experiments and detailed analyses to further clarify the technical details.
>
> ---
>
> ### W1: Regarding the impact of ParseKG on framework robustness
>
> Thanks for your valuable feedback. Existing graph construction methods rely on LLM-based entity and relation extraction, which suffer from unstable generation (Section 2.2, Case 1). Our ParseKG instead uses a spaCy-based NER model and avoids complex relation extraction, **ensuring consistent graph construction**. Following your suggestion, we add **stability analysis for graph construction**.
>
> *Table 1: F1 (3 Runs) of Different Graph Construction Methods on HotpotQA Bri.$^\text{2H}$*
>
> |Method|1st|2nd|3rd|Variance|
> |-|-|-|-|-|
> |LightRAG (OpenIE-based KG)|69.4|62.9|65.9|7.056|
> |KG-Translator (spaCy-based ParseKG)|70.9|71.9|70.4|0.389|
>
> As shown in Table 1, **ParseKG is more stable across three runs** (variance: 0.389 < 7.056), demonstrating stronger framework robustness.
>
> In addition, we verify that **ParseKG remains stable across 7 question types at different corpus scales (5k, 10k, 35k)**, with fluctuations within 2% (Appendix G.6), and **summarize its impact** in Appendix G.8.
>
> ---
>
> ### W2 & W3: Regarding further explanations on the translation paradigm and its components
>
> Thanks for your expertise. Existing structure-augmented RAG methods suffer from potential retrieval suspension and cumulative semantic drift (Section 2.2, Case 1&2), due to:
>
> + **Low-quality structures:** The absence of relations in the graph breaks search paths and prevents retrieval from reaching target knowledge.
>
> + **Inaccurate semantics:** Existing retrieval relies on semantic embeddings for query-structure matching, limited by insufficient text detail representation and noise from low-quality graphs.
>
> Motivated by this, we propose a novel translation paradigm that transforms retrieval into a translation task, translating queries into graph-level clues. Specifically,
>
> + We design **ParseKG**, which constructs a reliable, relation-extraction-free graph using lightweight NLP parsers to **comprehensively preserve knowledge associations and provide accurate entry points for queries**.
>
> + On top of ParseKG, we propose **KG-Translator**, which integrates query-passage matching into LLM inference via constrained decoding, **fully mining query-related semantics to offer precise retrieval guidance**.
>
> Furthermore, we present relevant experiments (excerpted from Table 2 and Appendix I.2 in the paper) to verify the sources of effectiveness.
>
> *Table 2: Ablation study on KG-Translator*
>
> |Variant|F1|
> |-|-|
> |KG-Translator|**68.50**|
> |*w/o* structure-aware fine-tuning|59.36|
> |*w/o* reverse translation|66.08|
> |*w* reverse-then-forward|59.79|
>
> As shown in Table 2, KG-Translator can **obtain precise graph-level retrieval clues** with our training strategy and translation mechanism.
>
> *Table 3: Query Intent Understanding of KG-Translator*
>
> |Type|HotpotQA|2WikiMQA|MuSiQue|
> |-|-|-|-|
> |Ambiguous intent|27.2%|38.7%|33.1%|
> |Unambiguous intent|72.8%|61.3%|66.9%|
>
> As shown in Table 3, KG-Translator can **identify ambiguous query intents** (e.g., abbreviations) and link them to ParseKG.
>
> *Table 4: Clue Quality of KG-Translator*
>
> |Distance from the Query|HotpotQA|2WikiMQA|MuSiQue|
> |-|-|-|-|
> |Passage|0.448|0.446|0.362|
> |Clue|0.478 (+6.7%)|0.498 (+11.7%)|0.423 (+17.0%)|
>
> As shown in Table 4, KG-Translator can generate clues **focusing on the core query semantics** with ParseKG.
>
> In summary, **translation paradigm unifies ParseKG's structural strengths and KG-Translator's translation ability**, with the two core components working synergistically.
>
> ---
>
> ### W4: Regarding the efficiency
>
> Thanks for your careful comments. We have provided an end-to-end latency analysis in Appendix G.2. Here, we further optimize the code via caching techniques and present **detailed efficiency analyses on the small-scale and large-scale corpora**.
>
> *Table 5: Component-wise Retrieval Time of KG-Translator*
>
> |Time (s)|2WikiMQA (5k)|ScaleQA (35k)|
> |-|-|-|
> |NER|0.067 (5.2%)|0.067 (2.8%)|
> |Entity-constrained Indexing (Trie)|0.455 (35.3%)|1.324 (56.0%)|
> |Entity Translation|0.173 (13.4%)|0.173 (7.3%)|
> |Relation-constrained Indexing (FM-Index)|0.021 (1.6%)|0.071 (3.0%)|
> |Relation Translation|0.556 (43.2%)|0.590 (24.9%)|
> |Ranking|0.016 (1.2%)|0.141 (5.9%)|
> |Total Time (s)|1.288|2.367|
>
> According to Table 5, scaling up the corpus brings **no significant changes to the latency of most components**.
>
> Due to space constraints, we provide **the full-stage efficiency comparison between KG-Translator and other baselines** in our **response to Reviewer 5AAG regarding W3**.
>
> ---
>
> Once again, we sincerely appreciate your time, efforts, and thoughtful feedback. If you have any further comments or recommendations, we would be grateful to receive them.
>
> Best regards!

---

> > ### Author Rebuttal · Reviewer_Rc27 · 2026-04-04
> >
> > Thank the authors for the responses. I decide to raise my score.

---

> > > ### Author Response · Authors · 2026-04-04
> > >
> > > Dear Reviewer Rc27,
> > >
> > > Thanks for your expertise and recognition of our response. Your constructive feedback has been invaluable in strengthening our submission, and we deeply appreciate your time and effort throughout the review process.
> > >
> > > Best regards,
> > >
> > > Authors of Paper 12261

---

### Official Review · Reviewer_RJuT · 2026-03-12

**Soundness:** 3
**Presentation:** 4
**Significance:** 3
**Originality:** 4
**Overall Recommendation:** 5
**Confidence:** 4

**Summary:**

This paper identifies two critical bottlenecks in existing structure-augmented RAG systems: potential retrieval suspension caused by missing implicit relations during OpenIE-based graph construction, and cumulative semantic drift where embedding-based matching propagates noise through "similar but irrelevant" graph neighborhoods. To address them, the authors propose KG-Translator, which reframes retrieval as a constrained translation task: a lightweight NLP model first constructs a hypergraph (ParseKG) without LLM involvement, then a small fine-tuned LLM translates user queries into graph-level clues via Trie- and FM-index-constrained decoding, and finally the clues are traced back to original passages for ranking and answer generation. Experiments on multiple multi-hop QA benchmarks show that KG-Translator significantly outperforms state-of-the-art baselines while reducing graph construction time by 56.5% and eliminating token costs entirely.

**Compliance With Llm Reviewing Policy:**

Affirmed.

**Final Justification:**

Thanks for the authors' response. I will keep my positive score for this paper.

**Key Questions For Authors:**

See W1.

**Limitations:**

Yes.

**Strengths And Weaknesses:**

**Strengths:**

S1. The paper is well written and well structured. It analyzes the failure cases of existing Graph-based RAG systems in detail, which are caused by potential retrieval suspension and cumulative semantic drift.

S2. The proposed Parsing-based KG Construction is technically sound and more efficient than LLM-based extraction methods used in most of Graph-based RAG methods, which is quite essential in real-world applications.

S3. The idea of Structure-constrained KG Translation is quite novel, which guarantees that the LLM only generates clues that actually exist in the ParseKG, ensuring structural faithfulness and preventing hallucinations during the clue generation phase.

S4. Extensive experiments demonstrate the impressive performance of KG-Translator. And these gains are achieved with drastically lower computational overhead.

**Weaknesses:**

W1. The implementation details mention that multi-hop retrieval is implemented "by chaining query decomposition", but the main text lacks a formal explanation of how this decomposition is conducted and interacts with the constrained translation process.

---

> ### Author Rebuttal · Authors · 2026-03-30
>
> Dear RJuT,
>
> Thanks for your positive feedback on our work. We sincerely appreciate your recognition of our contributions and the significance of the research problems we address. Your support further strengthens our confidence in the proposed approach.
>
> ---
>
> ### W1: Regarding the supplementary explanations on question decomposition
>
> Thank you for your expertise. It is important to analyze the question decomposition (QD) module of our framework. Following your suggestion, we provide supplementary explanations on question decomposition as follows:
>
> + For single-hop questions: We directly perform retrieval using KG-Translator with no need for question decomposition.
>
> + For multi-hop questions: Common decomposition approaches fall into two main categories: iterative methods and directed acyclic graph (DAG)-based methods. Specifically,
>   - Iterative methods alternate between retrieval and decomposition. After each retrieval step, the large language model judges whether the current information is sufficient; if not, it generates the next sub-question for deeper retrieval.
>
>   - DAG-based methods pre-plan all sub-questions before retrieval and perform retrieval in topological order.
>
>
> In our work, we adopt iterative decomposition with CoT, which enables flexible control over the iteration count and improves sub-question quality.
>
>
> Notably, **the decoupled design of KG-Translator enables it to adapt to diverse question decomposition methods**. We provide an additional performance comparison of representative question decomposition methods in the table below.
>
> *Table 1: Performance Comparison of Different Question Decomposition (QD) Methods on HotpotQA*
>
> | Method                    | Recall@2 | Recall@5 | EM   | F1   |
> | ------------------------- | -------- | -------- | ---- | ---- |
> | No QD                     | 65.0     | 72.0     | 62.7 | 70.7 |
> | Iterative QD              | 73.4     | 85.7     | 71.6 | 79.4 |
> | Iterative QD + CoT        | 74.4     | 86.3     | 72.1 | 80.3 |
> | Directed Acyclic Graph QD | 76.3     | 87.3     | 73.4 | 81.1 |
>
> As shown in Table 1, KG-Translator is highly compatible with various question decomposition methods, and stronger decomposition methods yield greater performance gains, demonstrating its substantial potential for further optimization.
>
> ---
>
> Thank you again sincerely for your time, efforts, and insightful feedback. If you have any additional questions or suggestions, we would be happy to discuss them.
>
> Best regards!

---

> > ### Author Rebuttal · Reviewer_RJuT · 2026-04-02
> >
> > Thanks for the authors' response. I will keep my positive score for this paper.

---

> > > ### Author Response · Authors · 2026-04-03
> > >
> > > Dear  Reviewer RJuT,
> > >
> > > Thank you for your positive feedback on our work. We sincerely appreciate your recognition of our contributions and the significance of the research problems we address. We will supplement the revised version with additional explanations and analyses.
> > >
> > > Best regards,
> > >
> > > Authors of Paper 12261

---

### Decision · Program_Chairs · 2026-04-30

**Decision:**

Accept (regular)

**Comment:**

After reviewing the four reviews, I recommend a Weak Accept for this submission. While the paper has clear merits—particularly its novel formulation of structure-augmented RAG as a constrained translation task, strong empirical gains over baselines, and significant reductions in graph construction cost—the reviewers raise several non-trivial concerns that temper enthusiasm. These include potential robustness issues with the parsing pipeline, difficulty in isolating the source of performance gains, a performance ceiling tied to ParseKG quality that may limit scaling, and incomplete efficiency comparisons (e.g., end-to-end latency). One reviewer also notes that the conceptual novelty is more about effective combination than a fundamentally new paradigm. Nonetheless, three out of four reviewers lean toward acceptance (one Accept, two Weak Accept), and the technical soundness, clear writing, and practical efficiency advantages are widely acknowledged. The weaknesses, while valid, appear addressable through additional analysis and experiments, and do not invalidate the core contribution. Therefore, the paper falls within the scope of a Weak Accept.